# Characteristics of an Ising-like Model with Ferromagnetic and Antiferromagnetic Interactions

**DOI:** 10.3390/e25101428

**Published:** 2023-10-09

**Authors:** Boris Kryzhanovsky, Vladislav Egorov, Leonid Litinskii

**Affiliations:** 1Scientific Research Institute for System Analysis RAS, Moscow 117218, Russia; kryzhanov@mail.ru (B.K.); rvladegorov@rambler.ru (V.E.); 2Independent Researcher, Haifa 2641418, Israel

**Keywords:** Ising model, free energy, critical exponents, antiferromagnetic, balanced system, effective number of the nearest neighbors, layered media

## Abstract

In the framework of mean field approximation, we consider a spin system consisting of two interacting sub-ensembles. The intra-ensemble interactions are ferromagnetic, while the inter-ensemble interactions are antiferromagnetic. We define the effective number of the nearest neighbors and show that if the two sub-ensembles have the same effective number of the nearest neighbors, the classical form of critical exponents (α=0, β=1/2, $\gamma =\gamma^{\prime }=1$, δ=3) gives way to the non-classical form (α=0, β=3/2, $\gamma =\gamma^{\prime }=0$, δ=1), and the scaling function changes simultaneously. We demonstrate that this system allows for two second-order phase transitions and two first-order phase transitions. We observe that an external magnetic field does not destroy the phase transitions but only shifts their critical points, allowing for control of the system’s parameters. We discuss the regime when the magnetization as a function of the magnetic field develops a low-magnetization plateau and show that the height of this plateau abruptly rises to the value of one when the magnetic field reaches a critical value. Our analytical results are supported by a Monte Carlo simulation of a three-dimensional layered model.

## 1. Introduction

Magnetic properties of multilayer superlattices have been subject to intensive theoretical and experimental study. Special interest is given to multilayer structures consisting of thin layers of various ferromagnetic and antiferromagnetic materials [1,2,3]. An example of such structures is designs with alternating ferromagnetic and non-ferromagnetic layers. In these structures, the thickness of the non-ferromagnetic layer can be chosen in such a way that the long-range exchange interaction between the ferromagnetic layers is antiferromagnetic [4]. This sort of structure has a giant magnetoresistance. Ferromagnetic layered structures with antiferromagnetic inter-layer interactions are theoretically investigated in [5,6] in the framework of the Heisenberg model. These studies reveal that this kind of system can have many distinct phases: ferromagnetic, antiferromagnetic, paramagnetic, and a whirling phase. Papers [7,8] consider the effect of weak interaction between two square Ising lattices on the phase transitions in this system. It is shown that if a structure has alternating layers of different ferromagnetic materials, there might exist a compensation temperature, which is a temperature below the critical value, at which the full magnetization of the lattice is zero. The conditions for the appearance of the compensation effect and the critical behavior of such systems near the compensation temperature are investigated in [9,10,11,12,13,14,15]. In recent years, great consideration has been given to the study of spin systems with competing ferromagnetic and antiferromagnetic interactions, or with interactions between different groups of spins [16,17,18,19]. This sort of system usually has a complex landscape of free energy containing many local minima separated from the global minimum by a deep potential barrier.

In the works [20,21,22,23,24], the authors investigated the spin-1 Ising models, where a spin has three possible states. They showed that in these models the critical temperature shifts with increasing crystal field and at the certain values of the field the transition order changes from the second to the first order. In the present paper, we demonstrate that such critical behavior is possible in the two-state spin systems.

This paper considers the mean field approximation for the simplest Ising-like model having not only the global minimum, but also a local minimum of energy. Spins are divided into two subgroups (sub-ensembles). The intra-group interaction of spins is ferromagnetic, while spins from different sub-ensembles can interact in both a ferromagnetic and antiferromagnetic manner. With antiferromagnetic inter-group interaction, the ground state is the ferromagnetic ordering within sub-ensembles and antiferromagnetic ordering among spins of different sub-ensembles. Yet, there is also a local energy minimum corresponding to a complete ferromagnetic order.

To compare the mean-field model characteristics with real systems, we carried out a Monte Carlo simulation of a 3D layered model with ferromagnetic intra-layer interactions and antiferromagnetic inter-group interactions. We took into account only the interactions between the nearest neighbors. This model corresponds to realistic ferromagnetic structures with giant magnetoresistance. Although it is expected that the results of the mean-field approach will differ from the results for real systems, it is well known that this approach approximates the 3D systems quite well. We found that the simulation results agree qualitatively with the predictions of our model. The agreement is expected to be better for lattices with larger dimensions.

The paper is organized as follows. State equations for the mean-field model are given in Section 2. Section 3 investigates the critical properties of the system and derives the values of the critical exponents. The effect of an external magnetic field on the temperature dependencies of magnetization is considered in Section 4. Section 5 compares the results of the Monte Carlo simulation with the analytical results obtained in the previous chapters. The discussion and our conclusions are given in Section 6. To conclude, we would like to mention that in our paper [25], for the first time, we used a similar technique. 

## 2. Basic Expressions

### 2.1. Equations of State and Critical Temperature

Let us consider a system of *N* spins with a connection matrix $\hat{J}$. The system is divided into two groups I and II holding $p_{1}N$ and $p_{2}N$ spins, respectively ($N=p_{1}N+p_{2}N$, $p_{1}+p_{2}=1$). The inter-spin connection in group I is defined by a quantity $J_{11}$, in group II by $J_{22}$, and cross-connections between the spins of group I and group II are defined by a quantity $J_{12}$:(1)$$J_{11}=\frac{q_{11}}{N},\;J_{12}=\frac{q_{12}}{N},\;J_{22}=\frac{q_{22}}{N}.$$For definiteness, in what follows, we assume that spin interaction within each group is of ferromagnetic nature, the interaction between spins from different groups is antiferromagnetic, and the direction of the external magnetic field H is positive:$$q_{11}\geq 0,\;q_{22}\geq 0,\;q_{12}\leq 0,\;H\geq 0.$$The case $q_{12}>0$ in the absence of the external field was investigated earlier in [26,27].

The Hamiltonian of this system can be written as
(2)$$E_{N}=-\left(J_{11}\sum_{i,j=1}^{Np_{1}}S_{i}S_{j}+J_{22}\sum_{i,j=1}^{Np_{2}}S_{i}^{\prime }S_{j}^{\prime }+J_{12}\sum_{i=1}^{Np_{1}}\sum_{j=1}^{Np_{2}}S_{i}S_{j}^{\prime }\right)-H\left(\sum_{i=1}^{Np_{1}}S_{i}+\sum_{j=1}^{Np_{2}}S_{j}^{\prime }\right),$$
where S and $S^{\prime }$ are the values of the spins from groups I and II, respectively. The sums in (2) are taken over all the spins of the corresponding groups. Introducing partial magnetizations of the two groups, $m_{1}$ and $m_{2}$: $$m_{1}=\frac{1}{Np_{1}}\sum_{i=1}^{Np_{1}}S_{i},\;m_{2}=\frac{1}{Np_{2}}\sum_{j=1}^{Np_{2}}S_{j}^{\prime },$$We can represent the energy of the system per one spin in the mean-field approximation, $E=E_{N}/N$, as follows:(3)$$E=-\frac{1}{2}\left(q_{11}p_{1}^{2}m_{1}^{2}+2q_{12}p_{1}p_{2}m_{1}m_{2}+q_{22}p_{2}^{2}m_{2}^{2}\right)-H(p_{1}m_{1}+p_{2}m_{2}),$$
and the full magnetization of the system becomes: $M=p_{1}m_{1}+p_{2}m_{2}$.

The statistical sum of this spin system is determined as:
(4)$$Z=\sum_{n_{1}=0}^{Np_{1}}\sum_{n_{2}=0}^{Np_{2}}\left(\begin{array}{c} Np_{1}\\ n_{1} \end{array}\right)\left(\begin{array}{c} Np_{2}\\ n_{2} \end{array}\right)e^{-NKE},$$
where K=1/T is the inverse temperature, $n_{1}$ and $n_{2}$ denote, respectively, the number of downward spins (−1) in groups I and II; $n_{1}$ and $n_{2}$ are related to partial magnetizations by the following expressions:$$n_{1}=Np_{1}\frac{1+m_{1}}{2},n_{2}=Np_{2}\frac{1+m_{2}}{2}.$$

If the total number of spins N is large, then summation in Formula (4) can be replaced by integration. Using Stirling’s formula to estimate binomial coefficients, we obtain
$$Z=\int_{-1}^{1}\int_{-1}^{1}e^{-NF(m_{1},m_{2})}dm_{1}dm_{2},$$
where
(5)$$F(m_{1},m_{2})=p_{1}S_{1}(m_{1})+p_{2}S_{2}(m_{2})+KE(m_{1},m_{2}),$$
and
$$S_{i}(m_{i})=\frac{1+m_{i}}{2}\ln\frac{1+m_{i}}{2}+\frac{1-m_{i}}{2}\ln\frac{1-m_{i}}{2},\;i=1,2.$$

Let us estimate this integral using the saddle point method. The equations for the saddle point ($\partial F/\partial m_{1}=0$, $\partial F/\partial m_{2}=0$) are
(6)$$\begin{array}{c} \frac{1}{2K}\ln\frac{1+m_{1}}{1-m_{1}}=q_{11}p_{1}m_{1}-q_{12}p_{2}m_{2}+H\\ \frac{1}{2K}\ln\frac{1+m_{2}}{1-m_{2}}=q_{22}p_{2}m_{2}-q_{12}p_{1}m_{1}+H. \end{array},$$These equations define the equilibrium values of the partial magnetizations $m_{1}$ and $m_{2}$ at a given temperature K. Substituting these values into expression (5), we obtain the free energy per spin F.

Let us determine the critical temperature of a phase transition, $K_{c}$. Assuming H=0 and $m_{1}\to 0$, ($K\to K_{c}$), in the case of $q_{12}=0$ from (6), we obtain an obvious result: a system split into two non-interacting subsystems has two critical points:(7)$$K_{c}^{\left(1\right)}=\frac{1}{q_{11}p_{1}},\;K_{c}^{\left(2\right)}=\frac{1}{q_{22}p_{2}}$$Expression (7) fully agrees with the mean-field theory [8] because in the notations of (1) and (2), the quantities $q_{11}p_{1}$ and $q_{22}p_{2}$ are the numbers of the nearest neighbors in groups I and II.

In the general case of $q_{12}\neq 0$, the system has one critical point, which we can find from (6):$$K_{c}=\frac{2}{q_{11}p_{1}+q_{22}p_{2}+\sqrt{{\left(q_{11}p_{1}-q_{22}p_{2}\right)}^{2}+4p_{1}p_{2}q_{12}^{2}}}$$

### 2.2. Energy Minima and Critical Value $H_{c}$

To proceed further, it is necessary to investigate the presence of energy minima to which the system can converge when the temperature approaches zero. It is easy to see that the energy (3) can be at its minimum only in configurations where $m_{1}=\pm 1$ and $m_{2}=\pm 1$. The stability of such states (the presence of the minimum) is only possible when two conditions are simultaneously met:(8)$$H_{1}m_{1}>0\;\mathrm{and}\;H_{2}m_{2}>0$$
where $H_{1}$ and $H_{2}$ are the local fields acting on the spins of groups I and II, respectively:(9)$$H_{1}=q_{11}p_{1}m_{1}+q_{12}p_{2}m_{2}+H,\;H_{2}=q_{22}p_{2}m_{2}+q_{12}p_{1}m_{1}+H.$$

When there is no external magnetic field (H=1), the unconditionally stable configurations are $S_{+-}=S(m_{1}=+1,m_{2}=-1)$ and $S_{-+}=S(m_{1}=-1,\;m_{2}=+1)$, because spins of both groups are directed along the local fields acting on them: $H_{1}m_{1}=q_{11}p_{1}+\left|q_{12}\right|p_{2}>0$ and $H_{2}m_{2}=q_{22}p_{2}+\left|q_{12}\right|p_{1}>0$. The stability conditions (8) for configurations $S_{++}=S(m_{1}=+1,\;m_{2}=+1)$ and $S_{--}=S(m_{1}=-1,\;m_{2}=-1)$ are $H_{1}m_{1}=q_{11}p_{1}-\left|q_{12}\right|p_{2}>0$ and $H_{2}m_{2}=q_{22}p_{2}-\left|q_{12}\right|p_{1}>0$. Simple calculations show that these configurations correspond to energy minima only when two conditions are met simultaneously:(10)$$q_{11}q_{22}>q_{12}^{2}\;\mathrm{and}\;\frac{\left|q_{12}\right|}{q_{11}+\left|q_{12}\right|}<p_{1}<\frac{q_{22}\;}{q_{22}+\left|q_{12}\right|}.$$It follows from (3) that if H=0, the configurations $S_{+-}$ and $S_{-+}$ correspond to a doubly degenerate global energy minimum, while the configurations $S_{++}$ and $S_{--}$ are local minima if (10) is met.

When H≠0, the picture becomes more complicated—the number of minima can vary from one to four. This situation is illustrated in Figure 1, which demonstrates how configuration energies change with the growing strength of the field H. We are interested in the energy values in the configurations $S_{++}$($m_{1}=1,\;m_{2}=1$), $S_{+-}$($m_{1}=1,\;m_{2}=-1$) and $S_{-+}$($m_{1}=-1,\;m_{2}=1$):$$E_{++}=-\frac{1}{2}\left(q_{11}p_{1}^{2}+q_{22}p_{2}^{2}\right)+\left|q_{12}\right|p_{1}p_{2}-H,$$
$$E_{+-}=-\frac{1}{2}\left(q_{11}p_{1}^{2}+q_{22}p_{2}^{2}\right)-\left|q_{12}\right|p_{1}p_{2}-H(p_{1}-p_{2}),$$
$$E_{-+}=-\frac{1}{2}\left(q_{11}p_{1}^{2}+q_{22}p_{2}^{2}\right)-\left|q_{12}\right|p_{1}p_{2}-H(p_{2}-p_{1}).$$The energy $E_{--}$ in the configuration $S_{--}$($m_{1}=-1,\;m_{2}=1$) is of no interest, because it is a minimum only when H is very small. 

When condition (10) is not met and the field H is small, the system has two energy minima corresponding to the antiferromagnetic configurations $S_{+-}$ and $S_{-+}$ (see Figure 1). When the field reaches the value $H_{++}=\max(\left|q_{12}\right|p_{1}-q_{22}p_{2},\left|q_{12}\right|p_{2}-q_{11}p_{1})$, the ferromagnetic configuration $S_{++}$ also becomes the energy minimum. The minima achieved in the configurations $S_{+-}$ and $S_{-+}$ disappear, when the field H is larger than the values $H_{+-}=q_{22}p_{2}+\left|q_{12}\right|p_{1}$ and $H_{-+}=q_{11}p_{1}+\left|q_{12}\right|p_{2}$, correspondingly. The only one minimum remains in the configuration $S_{++}$. If the condition (10) is met, for all values of the field H there is the minimum in the configuration $S_{++}$ and when $H<\min(q_{22}p_{2}-\left|q_{12}\right|p_{1},q_{11}p_{1}-\left|q_{12}\right|p_{2})$ there is the fourth minimum in the configuration $S_{--}$.

Let us introduce the critical value of the magnetic field as follows:(11)$$H_{c}=\left|q_{12}\right|p_{\max},\;p_{\max}=\max(p_{1},\;p_{2}).$$

Let $p_{1}>p_{2}$. Then, from comparing $E_{++}$, $E_{+-}$ and $E_{-+}$, we can conclude:when $H<H_{c}$, the antiferromagnetic configuration $m_{1}=1,\;m_{2}=-1$ is the ground state of the system, and the ferromagnetic configuration $m_{1}=1,\;m_{2}=1$ is its local minimum;when $H>H_{c}$, the ground state is the ferromagnetic state $m_{1}=1,\;m_{2}=1$, and the antiferromagnetic configuration $m_{1}=1,\;m_{2}=-1$ is its local minimum.

If $p_{1}<p_{2}$, we should swap indices 1 and 2 in these expressions. The aforesaid is summarized in Table 1.

### 2.3. Balanced System and “Critical” Value $q_{12}=q_{12}^{*}$

Further analysis reveals that the thermodynamic characteristics of balanced and unbalanced systems are considerably different. We call a system balanced if it has the effective number of spin “nearest neighbors” in the first sub-ensemble is equal to that in the second sub-ensemble. This equality can be expressed as:(12)$$q_{11}p_{1}+q_{12}p_{2}=q_{22}p_{2}+q_{12}p_{1}.$$Indeed, a spin from the first sub-ensemble has $q_{11}p_{1}$ neighbors from this same sub-ensemble and $q_{12}p_{2}$ neighbors from the other sub-ensemble. The same is true for a spin from the second sub-ensemble. We use the term “effective” number of neighbors because the neighbors with antiferromagnetic interaction make a negative contribution to this number.

The balance (12) is possible only when a certain relationship between the interaction parameters holds. For a symmetric system with $p_{1}=p_{2}$ and $q_{11}=q_{22}$, it follows from (12) that the system is balanced for any $q_{12}$. In a more general case, the balance condition (12) can be rewritten as $q_{12}=q_{12}^{*}$, where $q_{12}^{*}<0$ is a certain “critical” value of antiferromagnetic interaction:(13)$$q_{12}^{*}=\frac{q_{11}p_{1}-q_{22}p_{2}}{p_{1}-p_{2}},\;q_{12}^{*}<0.$$Below we will use notations $q_{12}=q_{12}^{*}$ or $q_{12}\neq q_{12}^{*}$ with $q_{12}^{*}$ defined in (13) to indicate whether a system is balanced or unbalanced.

Note that a system can be balanced only within a specific range of parameters $p_{1,2}$ and $q_{11},\;q_{22}$. Indeed, it follows from (13) that the condition $q_{12}^{*}<0$ can be written as:(14)$$q_{12}^{*}<0\;if\;\{\;\begin{array}{ccc} \frac{q_{22}}{q_{11}+q_{22}}\leq p_{1}<\frac{1}{2} & , & q_{11}\geq q_{22}\\ \frac{1}{2}<p_{1}\leq \frac{q_{22}}{q_{11}+q_{22}} & , & q_{11}\leq q_{22}. \end{array}$$Relationship (14) is illustrated as a phase diagram in Figure 2.

## 3. Critical Behavior

Let us introduce the relative temperature t as follows
$$t=\frac{T-T_{c}}{T_{c}}=\frac{K_{c}-K}{K}.$$We will consider the behavior of various physical quantities near the critical temperature, paying special attention to the difference between critical exponents in the general case ($q_{12}\neq q_{12}^{*}$) and critical exponents in the case of the “critical” antiferromagnetic interaction ($q_{12}=q_{12}^{*}$).

### 3.1. Spontaneous Magnetization and Critical Exponent β


The case $q_{12}\neq q_{12}^{*}$. Let us consider the equation of state (6) when H=0 and $m_{1,2}\to 0$. Let us expand it in small parameters $m_{1,2}$ with an accuracy up to the terms of the order of $m_{1,2}^{3}$ and t. Then, for partial magnetizations near the critical temperature ($K>K_{c}$), we obtain:
(15)$$m_{1}^{2}=-3tD_{1},\;m_{2}^{2}=-3tD_{2},$$
where
$$\begin{array}{l} D_{1}=\frac{p_{2}(1-K_{c}q_{22}p_{2})\left[2-K_{c}(q_{11}p_{1}+q_{22}p_{2})\right]}{p_{1}{\left(1-K_{c}q_{11}p_{1}\right)}^{2}+p_{2}{\left(1-K_{c}q_{22}p_{2}\right)}^{2}},\\ D_{2}=\frac{p_{1}(1-K_{c}q_{11}p_{1})\left[2-K_{c}(q_{11}p_{1}+q_{22}p_{2}\right]}{p_{1}{\left(1-K_{c}q_{11}p_{1}\right)}^{2}+p_{2}{\left(1-K_{c}q_{22}p_{2}\right)}^{2}}. \end{array}$$



The full spontaneous magnetization $M_{0}$ is defined as:(16)$$M_{0}=p_{1}m_{1}+p_{2}m_{2}=\pm \;\sqrt{-3t}\left(p_{1}\sqrt{D_{1}}-p_{2}\sqrt{D_{2}}\right).$$

Here, we take into account the fact that with H=0 and $q_{12}<0$, the inequality $m_{1}m_{2}<0$ holds.
2.The case $q_{12}=q_{12}^{*}$. Here, $p_{1}\sqrt{D_{1}}=p_{2}\sqrt{D_{2}}$ and the parenthesized expression from (16) become zero. It means that in the expressions resulting from the expansion of the equation of state (6), we must retain the terms of the order of $t^{2}$. Then, the spontaneous magnetization near the critical temperature is described as
(17)$$M_{0}=\pm \;{\left(-t\right)}^{\frac{3}{2}}\frac{\sqrt{3}p_{1}p_{2}(p_{1}-p_{2})}{K_{c}\left|q_{12}^{*}\right|{\left(1-3p_{1}p_{2}\right)}^{3/2}}.$$Thus, in an unbalanced system ($q_{12}\neq q_{12}^{*}$), the critical exponent β=1/2, which agrees with the conventional mean-field theory. In a balanced system ($q_{12}=q_{12}^{*}$), the critical exponent takes a “non-classical” value β=3/2. Here, by the classic value, we mean β=1/2 for the case where the interaction is ferromagnetic only.

### 3.2. Jump in Heat Capacity (H=0) and Critical Exponent α

At the critical point, the heat capacity C experiences a finite jump. Indeed, when t>0, the heat capacity C=0, and when $t\to 0^{-}$, the quantity C=−K dE/dK can be easily found using (3) and (23). Then, for the heat capacity jump at the critical point, we have:(18)$$\Delta C=\underset{t\to 0^{-}}{\lim}C=\frac{3}{2}K_{c}p_{1}p_{2}\frac{{\left(q_{11}p_{1}-q_{22}p_{2}\right)}^{2}+4p_{1}p_{2}q_{12}^{2}}{p_{1}{\left(1-K_{c}q_{11}p_{1}\right)}^{2}+p_{2}{\left(1-K_{c}q_{22}p_{2}\right)}^{2}}.$$

When $q_{12}=q_{12}^{*}$, expression (18) takes the form:$$\Delta C=\frac{3}{2K_{c}}\frac{p_{1}p_{2}}{p_{1}^{3}+p_{2}^{3}}.$$

Since the heat capacity tends to a finite value at the critical point, the classical definition of the critical exponent α loses sense. Therefore, we will use its alternative definition [28]:(19)$$F_{+}(K)-F_{-}(K)∼t^{2-\alpha }\;\mathrm{when}\;t\to 0,$$
where $F_{+}(K)$ and $F_{-}(K)$ are the free energies for $K<K_{c}$ and $K>K_{c}$, respectively. To compute the exponent α from (19), the functions $F_{+}(K)$ and $F_{-}(K)$ must be analytically extended to the K-axis beyond their domain. Using the expression for free energy (5) and taking into account the equation of state (6) and expressions for spontaneous magnetization (23), we obtain:(20)$$F_{+}(K)-F_{-}(K)=-\frac{3t^{2}}{4}\left[2K_{c}\left(q_{11}p_{1}^{2}D_{1}-2q_{12}p_{1}p_{2}\sqrt{D_{1}D_{2}}+q_{22}p_{2}^{2}D_{2}\right)+p_{1}D_{1}^{2}+p_{2}D_{2}^{2}\right].$$Comparing (20) with (19), we obtain α=0, which agrees with the classical mean-field model. Note that this result is independent of the magnitude of the antiferromagnetic interaction, i.e., it holds both for $q_{12}=q_{12}^{*}$ and $q_{12}\neq q_{12}^{*}$.

Typical curves of spontaneous partial magnetization $m_{1,2}(K)$ and energy variance $\sigma^{2}(K)=C(K)/K$ as functions of temperature without an external field are shown in Figure 3.

### 3.3. Susceptibility χ (H=0) and Critical Exponents γ and $\gamma^{\prime }$

Let us consider the behavior of the susceptibility of the system near the critical point, given that H=0. We define the full and partial susceptibilities as follows:$$\chi =\frac{\partial M(K,H)}{\partial H}=p_{1}\chi_{1}+p_{2}\chi_{2},\;\chi_{1}=\frac{\partial m_{1}(K,H)}{\partial H},\;\chi_{2}=\frac{\partial m_{2}(K,H)}{\partial H}.$$Differentiating the equations of state (6) with respect to H and solving the resulting equations for $\chi_{1}$ and $\chi_{2}$ in view of (15), we obtain the following expressions:(21)$$\{\begin{array}{ccc} \chi_{1}=\frac{t+\left[1-K_{c}p_{2}(q_{22}-q_{12})\right]}{t\;\sqrt{{\left(q_{11}p_{1}-q_{22}p_{2}\right)}^{2}+4p_{1}p_{1}q_{12}^{2}}} & & \\ & & if\;t>0\\ \chi_{2}=\frac{t+\left[1-K_{c}p_{1}(q_{11}-q_{12})\right]}{t\;\sqrt{{\left(q_{11}p_{1}-q_{22}p_{2}\right)}^{2}+4p_{1}p_{1}q_{12}^{2}}} & & \end{array},$$
and
(22)$$\{\begin{array}{ccc} \chi_{1}=-K_{c}\frac{\left[1-K_{c}p_{2}(q_{22}-c)]\;-\;\;t(1-3D_{2}\right)}{t\;\;\left[\left(1-K_{c}q_{11}p_{1})(1-3D_{1})+(1-K_{c}q_{22}p_{2})(1-3D_{2}\right)\right]} & & \\ & & if\;t<0\\ \chi_{2}=-K_{c}\frac{\left[1-K_{c}p_{1}(q_{11}-c)]\;-\;t(1-3D_{1}\right)}{t\;\;\left[\left(1-K_{c}q_{11}p_{1})(1-3D_{1})+(1-K_{c}q_{22}p_{2})(1-3D_{2}\right)\right]} & & \end{array},$$

When $q_{12}\neq q_{12}^{*}$, the terms in the square brackets in the numerators of (21) and (22) are not zero, and the quantity t in the numerators can be neglected. In this event, the susceptibility of the system $\chi =p_{1}\chi_{1}+p_{2}\chi_{2}$ takes the classical form:(23)$$\chi =\frac{1-K_{c}p_{1}p_{2}(q_{11}+q_{22}-2q_{12})}{\left|t\right|\sqrt{{\left(q_{11}p_{1}-q_{22}p_{2}\right)}^{2}+4p_{1}p_{1}q_{12}^{2}}}\;\;if\;\;t>0\;\;or\;\;t<0\;,$$
holding true for both t>0  and t<0 .When $q_{12}=q_{12}^{*}$, we have $\left[1-K_{c}p_{2}(q_{22}-q_{12}^{*})\right]=\left[1-K_{c}p_{1}(q_{11}-q_{12}^{*})\right]=0$. Then, $\chi_{1}$ and $\chi_{2}$ in (21) and (22) become constant values, and full susceptibility $\chi =p_{1}\chi_{1}+p_{2}\chi_{2}$ takes the same “non-classical” form for both t>0  and t<0 :(24)$$\chi =\frac{1}{\left|q_{12}^{*}\right|}.$$From (23) and (24), we can derive critical exponents:$$\{\begin{array}{cc} \gamma =\gamma^{\prime }=1 & \;if\;\;q_{12}\neq q_{12}^{*}\;\\ \gamma =\gamma^{\prime }=0 & if\;\;q_{12}=q_{12}^{*} \end{array}$$

It should be noted that in a balanced system (12) (at $q_{12}=q_{12}^{*}$), partial susceptibilities $\chi_{1}$ and $\chi_{2}$ experience a finite jump at the critical point; yet K-dependence of the full susceptibility χ remains continuous.

### 3.4. Scaling Hypothesis and Critical Exponent δ

According to the scaling hypothesis, the field H is a homogeneous function of variables $M^{1/\beta }$ and t near the critical point. Let us consider how the critical exponent δ and the form of the scaling function rely on the interaction parameters.
The case $q_{12}\neq q_{12}^{*}$ (unbalanced system). Expanding the equations of state (6) in terms of the small parameters $m_{1}$, $m_{2}$, t and extracting full magnetization $M=p_{1}m_{1}+p_{2}m_{2}$, we obtain:$$K_{c}H=M^{3}R_{1}+tMR_{2},$$
where
(25)$$R_{1}=\frac{{\left(1-K_{c}q_{11}p_{1}\right)}^{2}p_{1}+{\left(1-K_{c}q_{22}p_{2}\right)}^{2}p_{2}}{3p_{1}p_{2}{\left[1-K_{c}p_{1}p_{2}(q_{11}+q_{22}-2q_{12})\right]}^{2}},\;R_{2}=\frac{K_{c}\sqrt{{\left(q_{11}p_{1}-q_{22}p_{2}\right)}^{2}+4p_{1}p_{2}q_{12}^{2}}}{1-K_{c}p_{1}p_{2}(q_{11}+q_{22}-2q_{12})}.$$
It is easy to see that the scaling hypothesis in this case is confirmed, because if $q_{12}\neq q_{12}^{*}$, it follows from (16) that β=1/2. Indeed, expression (25) can be represented in the classical form $K_{c}H=M{\left|M\right|}^{\delta -1}h_{s}\left(t{\left|M\right|}^{-1/\beta }\right)$ with the critical exponent δ=3 and the scaling function $h_{s}(x)$ of the form:$$h_{s}(x)=R_{1}+R_{2}x,\;\mathrm{where}\;x=t/M^{2}.$$The case $q_{12}=q_{12}^{*}$ (balanced system (12)). Performing similar calculations with regard to the relations $1-K_{c}p_{1}\left(q_{11}+\left|q_{12}^{*}\right|\right)=1-K_{c}p_{2}\left(q_{22}+\left|q_{12}^{*}\right|\right)=0$, which take place when $q_{12}=q_{12}^{*}$, we obtain:(26)$$K_{c}H=\{\begin{array}{lll} K_{c}\left|q_{12}^{*}\right|M & , & t>0\\ K_{c}\left|q_{12}^{*}\right|M-{\left|t\right|}^{3/2}R_{3} & , & t<0 \end{array}\;,\;\mathrm{where}\;R_{3}=\frac{\sqrt{3}\;p_{1}p_{2}\left|p_{1}-p_{2}\right|}{{\left(1-3p_{1}p_{2}\right)}^{3/2}}.$$
In this event, (17) gives β=3/2. Correspondingly, expression (26) can be rewritten in the classical form $K_{c}H=M{\left|M\right|}^{\delta -1}h_{s}\left(t{\left|M\right|}^{-1/\beta }\right)$, with the critical exponent δ=1 and the scaling function $h_{s}(x)$ of the form:$$h_{s}(x)=\{\;\begin{array}{lll} K_{c}\left|q_{12}^{*}\right| & , & t>0\\ K_{c}\left|q_{12}^{*}\right|-{\left|x\right|}^{3/2}R_{3} & , & t<0 \end{array},\;\mathrm{where}\;x=t/M.$$
It is easy to see that with $q_{12}=q_{12}^{*}$, the Rushbrooke relation, α+2β+γ≥2, the Widom relation, γ≥β(δ−1), and the Griffiths relation, β(δ+1)≥2−α, hold as strict inequalities. At the same time, the monograph [28] claims that, being a consequence of the scaling hypothesis, these relations should turn into strict equalities. Thus, the validity of the scaling hypothesis remains in question when $q_{12}=q_{12}^{*}$.


Summing up the above, we should note that the critical exponents of the model under consideration correspond to the parameters of the classical mean-field model provided that $q_{12}\neq q_{12}^{*}$, i.e., when the effective numbers of neighbors in different sub-lattices are not equal (see Table 2). However, in a balanced system, when condition (12) is met (i.e., $q_{12}=q_{12}^{*}$), the critical exponents β, γ and δ take “non-classical” values. Then, the relation α+2β+γ=2, which is a consequence of the scaling hypothesis, is violated.

At the end of this section let us show how functions of certain physical quantities change when $q_{12}=q_{12}^{*}$. Figure 4a presents the temperature dependence of susceptibility in the absence of an external field χ(K,0). We can see that when $q_{12}\neq q_{12}^{*}$, the susceptibility diverges at the critical point, just as in the case of the classical mean-field model. When $q_{12}=q_{12}^{*}$, the susceptibility has a finite critical value defined by expression (24). At the “critical” values of antiferromagnetic interactions and critical temperature, the field-dependences of partial magnetizations $m_{1}(K_{c},H)$ and $m_{2}(K_{c},H)$ also change (Figure 4b). If $q_{12}\neq q_{12}^{*}$ and the field H is small, the system is in the antiferromagnetic state ($m_{1}m_{2}<0$), and it makes a cross-over to the ferromagnetic state ($m_{1}m_{2}>0$) when the field increases. If $q_{12}=q_{12}^{*}$, the system is in the ferromagnetic state at any value of H, and its partial magnetizations are equal ($m_{1}=m_{2}$). 

## 4. Dependences of Physical Quantities on Temperature and External Field

In this section, we investigate how the physical parameters of the system depend on temperature in the presence of an external field. We find that these dependences change dramatically when the antiferromagnetic interaction parameter $q_{12}$ takes a “critical” value $q_{12}=q_{12}^{*}$. In the first subsection, we deal with the situation when $q_{12}\neq q_{12}^{*}$. In the second subsection, we consider a symmetric configuration $q_{11}=q_{22},$ $p_{1}=p_{2}=1/2$, which is a particular case of a system with “critical” antiferromagnetic interaction.

### 4.1. Unbalanced System ($q_{12}\neq q_{12}^{*}$)

As follows from the equations of state (6), at $H\geq H_{c}$ the right sides of both equations of state are necessarily positive and, therefore, when $m_{1,2}>0$ the system is in the ferromagnetic phase at any temperature. That is why the change in phase is only possible when the external field is weak, $H<H_{c}$. It follows from expression (9) for the local fields that for small K the local fields $H_{1,2}>0$ and, therefore, when $m_{1,2}>0$, both sub-groups are in the ferromagnetic phase. If $H<H_{c}$, when $m_{1}m_{2}<0$, there is a temperature $K_{p}(H)$, at which point the transition into the antiferromagnetic phase occurs. If $q_{12}\neq q_{12}^{*}$, there are four fundamentally different behaviors of partial magnetizations as functions of temperature, $m_{1,2}(K)$, as determined by the characteristics of the model and by the strength of the external field H (Figure 5 and Figure 6).

Before we proceed to further analysis, let us consider the behavior of partial magnetizations at high temperatures, i.e., at K→0. Differentiating Equation (6) with respect to K and retaining only the quantities of the first order of smallness, we obtain the following expressions for the derivatives ${\dot{m}}_{1}$ and ${\dot{m}}_{2}$ at K→0:(27)$$\begin{array}{ccc} {\dot{m}}_{1}>{\dot{m}}_{2} & if & p_{1}\left(q_{11}+\left|q_{12}\right|\right)>p_{2}\left(q_{22}+\left|q_{12}\right|\right),\\ {\dot{m}}_{2}>{\dot{m}}_{1} & if & p_{1}\left(q_{11}+\left|q_{12}\right|\right)<p_{2}\left(q_{22}+\left|q_{12}\right|\right). \end{array}$$This implies that with growing K, the partial magnetization $m_{1}$ increases faster than $m_{2}$ if the effective number of neighbors in the lattice I is greater than that in the lattice II, and vice versa. For definiteness, in this subsection, we will assume that $p_{1}\left(q_{11}+\left|q_{12}\right|\right)>p_{2}\left(q_{22}+\left|q_{12}\right|\right)$. If the opposite is true, all considerations will hold up to a permutation of the indices 1 and 2.

1.The case $H<H_{c}$. Let us first consider the case $H<H_{c}$, when the ground state is antiferromagnetic ($m_{1}m_{2}=-1$ at K→∞).

Let $p_{1}\left(q_{11}+\left|q_{12}\right|\right)>p_{2}\left(q_{22}+\left|q_{12}\right|\right)$. Then, in accordance with (27), for K<<1, the magnetization $m_{1}$ grows with K faster than $m_{2}$. In this case, the magnetization $m_{1}$ will keep growing as K increases, while the magnetization $m_{2}$ will initially grow and then will drop to negative values, passing through a zero value at a particular temperature $K=K_{p}$. At further growth of K, there are two distinct scenarios for the curve development in the $K>K_{p}$ region: If $p_{1}>p_{2}$, the partial magnetization $m_{1}$ will monotonously increase, and the partial magnetization $m_{2}$ will decrease, tending to 1 and −1, correspondingly (Figure 5a). There is no phase transition in this case.If $p_{1}<p_{2}$, the system’s ground state is the configuration with $m_{1}=-1$, $m_{2}=1$. At a certain temperature, a first-order phase transition will occur, as a result of which the magnetization $m_{1}$ will become negative, and the magnetization $m_{2}$ will be positive (Figure 5b). This transition will be accompanied by an abrupt change in the measurable parameters: magnetization M(K), internal energy U(K), and heat capacity C(K).

Note that the full magnetization of the system $M=p_{1}m_{1}+p_{2}m_{2}$ always remains positive, and in the limit K→∞, we have $M=\left|p_{1}-p_{2}\right|$ independently of the relationship between $p_{1}$ and $p_{2}$.

The temperature $K_{p}$ of the phase change, i.e., the temperature at which $m_{2}$ passes zero, can be easily found: at $m_{2}=0$, the second part of Equation (6) reduces to $\left|q_{12}\right|p_{1}m_{1}=H$. Substituting this relation in the first part of Equation (6), we obtain:(28)$$K_{p}=\frac{\left|q_{12}\right|}{2H(q_{11}+\left|q_{12}\right|)}\;\ln\frac{p_{1}\left|q_{12}\right|+H}{p_{1}\left|q_{12}\right|-H},\;m_{2}(K=K_{p})=0,\;m_{1}(K=K_{p})=\frac{H}{\left|q_{12}\right|p_{1}}.$$

When the external field is weak ($H\ll p_{1}\left|q_{12}\right|$), the temperature of the transition into the antiferromagnetic phase approaches the quantity
(29)$$K_{p}(H\to 0)=\frac{1}{p_{1}(q_{11}+\left|q_{12}\right|)}$$

Let us compare (29) with the temperature $K_{c}$. It is easy to see that inequality $K_{p}(H\to 0)\leq K_{c}$ is identical to condition (27), i.e., when the field is weak, the temperature $K_{p}$ always tends to a value less or equal to $K_{c}$. The two temperatures can be equal, $K_{p}(H\to 0)=K_{c}$, only if $q_{12}=q_{12}^{*}$. 

Moreover, the expression (28) is correct given that $p_{1}\left|q_{12}\right|>H$. It means that $m_{2}$ can pass zero only if the external field is relatively weak, or if the transition into the antiferromagnetic phase is an abrupt process. Otherwise, $m_{2}$ cannot pass zero, and $m_{1,2}\left(K\right)\geq 0$ for any K.

2.If $H>H_{c}$, the behavior of the temperature dependences is quite predictable, without any peculiarities (see Figure 6). The partial magnetizations are positive at any K. In particular, if the field H is close enough to the critical value $H_{c}$, the magnetization $m_{2}$ decreases with the growth of K in a certain temperature range, but it never becomes negative (Figure 6a). When $H\gg H_{c}$, the partial magnetizations $m_{1}$ and $m_{2}$ are increasing functions of K over the whole temperature range. When $H>H_{c}$, the ground state of the system is the ferromagnetic state: $M=m_{1}=m_{2}=1$, K→∞.

Summarizing the results of Section 4.1, we can highlight the following properties of an unbalanced ($q_{12}\neq q_{12}^{*}$) system: Firstly, the presence of magnetic field suppresses a second-order phase transition in the system. Secondly, if $0<H<H_{c}$, a first-order phase transition can occur for a certain relation between parameters (27) and $p_{1,2}$, as illustrated in Figure 5b.

### 4.2. Balanced System ($q_{12}=q_{12}^{*}$)

If the condition of equal effective numbers of neighbors (12) is met, there is a solution to the equations of state (6): $m_{1}=m_{2}$. This solution does not always correspond to the free energy minimum, and the behavior of partial magnetizations is so diverse that the analysis of the general case ($q_{11}\neq q_{22}$,$p_{1}\neq p_{2}$) is not within the scope of this paper.

Here, we consider the simplest (symmetrical) case, when
(30)$$q_{11}=q_{22}=q\;\mathrm{and}\;p_{1}=p_{2}=\frac{1}{2}.$$

Note that even in this simplest case, the behavior of partial magnetizations can be fairly diverse (Figure 7). Although the below formulae are true only for symmetric systems (30), general patterns hold true for all systems with $q_{12}=q_{12}^{*}$.

We recall that in the symmetric model, the equality $q_{12}=q_{12}^{*}$ is true for all negative $q_{12}$. The critical temperature in this case is determined as:$$K_{c}=\frac{2}{q+\left|q_{12}\right|},$$
and equations of state (6) take the form:(31)$$\begin{array}{c} \frac{1}{K}\ln\frac{1+m_{1}}{1-m_{1}}=qm_{1}-\left|q_{12}\right|m_{2}+2H,\\ \frac{1}{K}\ln\frac{1+m_{2}}{1-m_{2}}=qm_{2}-\left|q_{12}\right|m_{1}+2H. \end{array}$$

As we see, these equations have the same coefficients and admit a solution $m_{1}=m_{2}$. Indeed, analysis of (31) shows that over a certain temperature interval $0\leq K\leq K_{S}$, partial magnetizations $m_{1}=m_{2}=m$ are defined by the equation:(32)$$\frac{1}{K}\ln\frac{1+m}{1-m}=m\left(q-\left|q_{12}\right|\right)+2H$$Magnetization m is positive because the external magnetic field H is positive. If the field is smaller than a certain value $H_{s\max}$, at a temperature $K=K_{s}$, the curve m=m(K) experiences a “soft” splitting into two diverging partial magnetizations $m_{1}(K)$ and $m_{2}(K)$ (Figure 7a).
Soft splitting and merging. Let us determine the “soft” splitting point $K_{S}$. Quantities $m_{1}=m_{2}=m_{3}$ at this point are derived from (32) at $K=K_{S}$. Let us consider the behavior of magnetizations $m_{1,}{}_{2}$ in a small vicinity of $K_{S}$, introducing the small deviation
$$t_{s}=\frac{T-T_{s}}{T_{s}}=\frac{K_{s}-K}{K},\;\left|t_{s}\right|\to 0.$$

In this case, we seek partial magnetizations in the form $m_{1}=m_{2}=m_{S}-\delta$ when $K<K_{s}$ and in the form $m_{1,2}=m_{S}\pm \;\;\delta_{1,2}$ when $K>K_{s}$, where δ→0 and $\delta_{1,}{}_{2}\to 0$. Let us substitute $m_{1,2}$ of this form into (31) and (32) and carry out expansion in small parameters δ, $\delta_{1,}{}_{2}$ retaining terms up to $\delta_{1,2}^{3}$. Equating the terms of the same order of smallness, we obtain the expression for the splitting point:(33)$$K_{s}=\frac{K_{c}}{1-m_{s}^{2}},$$
as well as the expressions for small deviations from $m_{S}$:(34)$$\begin{array}{l} \delta =t_{s}\kappa ,\\ \delta_{1,}{}_{2}=\sqrt{-t_{s}\kappa_{1}}\pm \kappa_{2}t_{s}, \end{array}$$
where
(35)$$\begin{array}{l} \kappa =\frac{\left(q-\left|q_{12}\right|\right)m_{s}+2H}{2\left|q_{12}\right|}\\ \kappa_{1}=3\left(1-m_{s}^{2}\right)\left(\frac{\left|q_{12}\right|-qm_{s}^{2}-2Hm_{s}}{\left|q_{12}\right|-3qm_{s}^{2}}\right)\\ \kappa_{2}=\frac{(1+3m_{s}^{2})\kappa_{1}-3{\left(1-m_{s}^{2}\right)}^{2}}{6m_{s}(1-m_{s}^{2})} \end{array}$$

2.Splitting point and “bubble” formation. The quantity $m_{S}$ is a solution to Equation (32) at$K=K_{S}$. This equation allows for a simple graphical solution if we rewrite it in the form:

(36)$$H=R(m_{s})\equiv \frac{1}{2}m_{s}\left(\left|q_{12}\right|-q\right)+\frac{1}{4}\left(\left|q_{12}\right|+q\right)(1-m_{s}^{2})\ln\frac{1+m_{s}}{1-m_{s}}$$One example of such a solution for $\left|q_{12}\right|<q$, $\left|q_{12}\right|=q$ and $\left|q_{12}\right|>q$ is given in Figure 8. The function $R(m_{s})$ has one maximum at $m_{s}=m_{\max}$, where $m_{\max}$ is the solution to the equation ∂R(m)/∂m=0:(37)$$m_{\max}\ln\frac{1+m_{\max}}{1-m_{\max}}=\frac{2\left|q_{12}\right|}{q+\left|q_{12}\right|}.$$Correspondingly, Equation (36) can have a solution only when $H\leq H_{\max}$, where
(38)$$H_{\max}=\frac{\left(\left|q_{12}\right|-qm_{\max}^{2}\right)}{2m_{\max}}.$$

If $\left|q_{12}\right|=q$, the right wing of the curve $R(m_{s})$ is on the abscissa axis. In the case $\left|q_{12}\right|<q$, the curve is lower than the abscissa. As one can see, if $\left|q_{12}\right|\leq q$, Equation (36) may have two solutions. The first one ($m_{s}^{\prime }\leq m_{\max}$) corresponds to the splitting point of the curves $m_{1}(K)$ and $m_{2}(K)$, the second solution ($m_{s}^{\prime \prime }\geq m_{\max}$) corresponds to the merging point, and the formation of a “bubble” is shown in Figure 7b,c. 

If $\left|q_{12}\right|>q$, the right wing of the curve $R(m_{s})$ behaves as $R=\left(\left|q_{12}\right|-q\right)/2$ and is above the abscissa. Therefore, for $0<H\leq \left(\left|q_{12}\right|-q\right)/2$, Equation (36) has only one solution ($m_{s}\leq m_{\max}$), which is responsible for the splitting, whereas for $\left(\left|q_{12}\right|-q\right)/2<H\leq H_{\max}$, there are two solutions $m_{s}^{\prime }\leq m_{\max}$ and $m_{s}^{\prime \prime }\geq m_{\max}$, i.e., a “bubble” can form (as in Figure 7b,c).

Let us determine the asymptotic values of $m_{\max}$ and $H_{\max}$. In the case of weak antiferromagnetic interaction ($\left|q_{12}\right|<<q$), it follows from (37) and (38) that
(39)$$m_{\max}≃{\left|\frac{q_{12}}{q}\right|}^{\;1/2},\;H_{\max}≃\frac{{\left|q_{12}\right|}^{\;3/2}}{2\sqrt{q}},$$
and in the limit case $\left|q_{12}\right|>>q$, we have:(40)$$m_{\max}≃0.8336,\;H_{\max}=\frac{\left|q_{12}\right|}{\;2\;m_{\max}}≃1.2H_{c}.$$As we see, a soft splitting can be observed when H is varied within a wide enough range. Note that the width of this range grows with $\left|q_{12}\right|$.

If soft splitting is not feasible and the antiferromagnetic state ($H<H_{c}$) is the ground state, then passing through the critical point is accompanied by a jump of partial magnetizations (Figure 7b). If the condition of a “bubble” formation is met, a jump in magnetization occurs after the merging point at $H<H_{c}$ (Figure 7c).

Generally, the temperature behavior of magnetizations is very diverse (Figure 7), depending strongly on the relative magnitudes of the ratio of $\left|q_{12}\right|/q$ and H. This aspect of the model will be investigated in more detail in future papers.

3.Field-controlled phase transition at $K=K_{s}$. A second-order phase transition occurs at the point of soft splitting/merging and is accompanied by a jump in heat capacity. The magnitude of the heat capacity jump, Δ*C* under the splitting can be easily calculated by differentiating expression (3) for the energy, with the account of expression (35):

(41)$$\Delta C=\frac{3\left(q+\left|q_{12}\right|\right){\left(\left|q_{12}\right|-qm_{s}^{2}-2Hm_{s}\right)}^{2}}{4\left|q_{12}\right|\left(\left|q_{12}\right|-3qm_{s}^{2}\right)}.$$At the point of merging of the curves $m_{1}(K)$ and $m_{2}(K)$, ΔC is determined by the same expression, but with a different value of $m_{s}$.

It follows from what has been said that $K_{s}$ determined by (33) is nothing else but a magnetic-field-dependent critical point: $K_{s}=K_{c}(H)$. In other words, in a symmetrical system, the external field does not suppress a phase transition but only shifts it towards larger temperatures: $K_{c}(H\neq 0)>K_{c}(H=0)\equiv K_{c}$. The range of the critical point values can be quite large. As follows from (33), given strong antiferromagnetic interaction ($\left|q_{12}\right|>>q$) the phase transition at the soft-splitting point can be greatly shifted: with the field changing from H=0 to $H=H_{\max}\sim 1.2H_{c}$ ($m_{s}$ changing from $m_{s}=0$ to $m_{s}=m_{\max}\sim 0.83$), $K_{s}$ changes from $K_{s}=K_{c}$ to $K_{s}≃3.3\;K_{c}$. The variations at the merging point (if any) cover a yet wider range, where the case of $m_{s}\to 1$, (i.e., *K*_s_ →∞) can take place. For instance, Figure 7f demonstrates the case when, with $H≃0.999\;H_{c}$, a “bubble” forms, and the system has two critical points: one at the splitting point at $K_{s}≃1.4\;K_{c}$ and the second at the merging point at $K_{s}≃8.3\;K_{c}$. We have deliberately chosen such interaction parameters that reveal the richness of the system: besides two second-order phase transitions, it has two first-order phase transitions (the curve $m_{2}(K)$ undergoes two abrupt changes).

4.Critical exponents of a phase transition at the point of soft splitting. Let us consider the critical behavior of the involved physical quantities near the temperature $K_{s}$.

We start by pointing out that the exponent $\alpha_{s}=0$, since, in accordance with Equation (41), at the splitting point the heat capacity experiences a finite jump. We will define other critical exponents as follows: $$m_{s}-M(K)∼{\left(-t_{s}\right)}^{\beta_{s}}\;\mathrm{with}\;t_{s}\to 0^{-},$$
$$m_{s}-M(K)∼t_{s}{}^{\beta_{s}^{\prime }}\;\mathrm{with}\;t_{s}\to 0^{+},$$
$$\chi (K)∼{\left(-t_{s}\right)}^{-\gamma_{s}}\;\mathrm{with}\;t_{s}\to 0^{-},$$
$$\chi (K)∼t_{s}{}^{-\gamma_{s}^{\prime }}\;\mathrm{with}\;t_{s}\to 0^{+},$$
$$M(K_{s},H)-m_{s}∼{\left(H-H_{s}\right)}^{1/\delta_{s}}\;\mathrm{with}\;H-H_{s}\to 0^{+},$$
$$m_{s}-M(K_{s},H)∼{\left(H_{s}-H\right)}^{1/\delta_{s}^{\prime }}\;\mathrm{with}\;H-H_{s}\to 0^{-},$$
where $H_{s}$ is the magnitude of the external field at which the phase transition happens.

It follows from Equation (34) that in the vicinity of the splitting point, the full magnetisation is determined by the following equations: (42)$$M=m_{s}+\kappa_{2}t_{s},\;t_{s}<0,\;M=m_{s}-\kappa t_{s},\;t_{s}>0.$$It follows from (42) that the critical exponents $\beta_{s}=\beta_{s}^{\prime }=1$. For H→0 ($m_{s}\to 0$), we have $\kappa_{2}\to 0$, and hence we should take into account the terms of the order of $t_{s}^{3/2}$ in the expansion of the full magnetization M over $t_{s}$. This agrees with the results obtained earlier in Section 3.1. We also point out that at large values of $\left|q_{12}\right|$, there is such a value of the magnetic field $H^{\prime }<H_{\max}$ beyond which $\kappa_{2}$ becomes negative, so that the magnetization increases when the temperature rises above the splitting point. 

We differentiate the equations of states (31) and (32) over H and, making use of Equations (33) and (34), we obtain the following for the susceptibilities at the critical points:(43)$$\begin{array}{l} \chi =\frac{1}{\left|q_{12}\right|},t_{s}\to 0^{+}\\ \chi ={\left[\left|q_{12}\right|+\frac{2\kappa_{1}m_{s}^{2}(q+\left|q_{12}\right|)}{\left(1-m_{s}^{2}+2m_{s}\right)\left(1-m_{s}^{2}\right)-(3m_{s}^{2}+1)\kappa_{1}}\right]}^{-1},t_{s}\to 0^{-} \end{array}$$It follows from (43) that the susceptibility has a finite jump in the critical point. Its magnitude depends on $m_{s}$, and hence on the magnitude of the field H. In the absence of the external field, the jump disappears. From Equation (43), we find the critical exponents $\gamma_{s}=\gamma_{s}^{\prime }=0$.

We will use Equation (32) to estimate the exponent $\delta_{s}$. Let us look at how the magnetization changes in response to a small increase in the external field. We introduce the notations $\Delta M=M-m_{s}$ and $\Delta H=H-H_{s}$, and accounting only for the first-order terms in ΔM, we find at the temperature $K_{s}$:(44)$$\Delta H=\left|q_{12}\right|\Delta M.$$Equation (44) is valid also for negative values of ΔH and ΔM; therefore, we obtain for the critical exponents: $\delta_{s}=\delta_{s}^{\prime }=1$.

We see that in this case, the relationships α+2β+γ=2, γ=β(δ−1) and β(δ+1)=2−α, which are the consequences of the scaling hypothesis, are satisfied in the form of equalities. Except for β, all other critical exponents at the soft splitting point are equal to their values in the absence of the external field. 

5.Restrictions in the soft splitting. Above we analyzed the processes of soft splitting and merging with the assumption that these regimes could be reached. However, sometimes this is not possible.

Firstly, the field H should be such that the condition $K_{s}<K_{p}$ is met, i.e., the curves $m_{1,2}(K)$ must split before one of them passes zero.

Secondly, the field cannot be too big, $H<H_{\max}$, otherwise the soft splitting will be replaced by an abrupt jump-like change in partial magnetizations $m_{1,2}(K)$.

Thirdly, the splitting at point Ks is possible if δ1 and δ2 take real values. It follows from (35) that δ1,δ2∈ℝ at ts<0, if
(45)(|q12|−qms2−2Hms)(|q12|−3qms2)>0.Expression (45) is a necessary condition for the soft splitting point to exist. The quantities δ1 and δ2 must be real numbers at ts>0 for Ks to be the point of merging. Correspondingly, the necessary condition for the point of merging to exist is as follows:(46)(|q12|−qms2−2Hms)(|q12|−3qms2)<0.Examination of (46) in light of (36)–(40) shows that merging of the curves m1(K) and m2(K) (i.e., the formation of a “bubble”) usually takes place only when the magnitude of H is very close to Hc.

## 5. Computer Simulation of the Layered Model

Let us compare the results of the mean-field model with the results of the computer simulation of lattices with a finite interaction radius. We want to make sure that in the case of “critical” antiferromagnetic interaction $q_{12}=q_{12}^{*}$, which takes place if the condition (12) is satisfied, the critical parameters indeed take non-classical values presented in Table 2. To this end, let us consider a three-dimensional cubic lattice consisting of alternating two-dimensional layers of spins (see Figure 9). The intra-layer spin interaction is described by the interaction constants $J_{11}>0$ and $J_{22}>0$ for even and odd layers, respectively. The interlayer interaction is antiferromagnetic, and it is described by the constant $J_{12}<0$. The interaction only between the nearest neighbors is taken into account. 

This model is described by the Hamiltonian
$$E_{N}=-\left(J_{11}\sum_{\langle i,j\rangle }S_{i}S_{j}+J_{22}\sum_{\langle i,j\rangle }{S^{\prime }}_{i}S_{j}^{\prime }+J_{12}\sum_{\langle i,j\rangle }S_{i}S_{j}^{\prime }\right)-H\left(\sum_{i=1}^{N_{1}}S_{i}+\sum_{j=1}^{N_{2}}S_{j}^{\prime }\right),$$
where 〈i,j〉 stands for a set of nearest neighbors. The relationship between the interaction parameters of the layered model and the parameters of the mean-field model is defined by the expression:$$q_{11}=8J_{11},\;q_{22}=8J_{22},\;q_{12}=4J_{12}.$$

Condition (12), under which $q_{12}=q_{12}^{*}$, is set for the simplest case of $p_{1}=p_{2}$, $q_{11}=q_{22}$.

The Metropolis algorithm is used to calculate the thermodynamic parameters for the lattices of the size N=L×L×L, where L is the dimension of the lattice varying from 6 to 64. The linear dimension L is always an even number to secure the equality $p_{1}=p_{2}$. The case of $p_{1}\neq p_{2}$ was also modeled preliminarily, but the results of the modeling showed that in this case, it is impossible to achieve the equality of the moduli of partial magnetizations, $\left|m_{1}\right|=\left|m_{2}\right|$, over a wide temperature range even if condition (12), the effective neighbor numbers being equal, is met. A possible cause for this is the effect of boundaries. Therefore, to make it possible to study “critical” antiferromagnetic interactions occurring in the balanced model ($q_{12}=q_{12}^{*}$), the modeling is carried out for the case of $p_{1}=p_{2}$ only.

1.Figure 10 shows the temperature dependences of partial magnetizations of the layered model when $q_{11}=q_{22}$. The general behavior of these dependences is similar to the dependences obtained in the mean-field model for a balanced system (c). Depending on the magnitude of the external field, we can observe both “soft” (Figure 10b) and “hard” (Figure 10c) splitting. However, we failed to observe magnetization “bubbles” in this model.

When the lattice dimension is big enough and the external field H is close to, but less than, the critical magnitude $H_{c}$, the system does not proceed into the antiferromagnetic phase by cooling; instead, it stays in the local energy minimum (Figure 10d). We observe the same situation for a fully connected lattice (the analog of the mean-field model) if we use the Metropolis algorithm to compute the thermodynamic parameters.

To avoid misunderstanding, we should add that “hard” splitting accompanied by an abrupt change in magnetization (shown in Figure 10c) was observed only with relatively small lattice dimensions (L≤24). A small lattice size makes it possible to overcome the energy barrier between the local and global minima. When dimensions are large (L=32, 64) and K→∞, the system always transits into the ferromagnetic state with the magnetization M=1, which is a local minimum at $H<H_{c}$.

2.Let us consider the temperature dependence of the susceptibility χ=χ(K) in the layered model in the absence of the external field: H=0. If $q_{11}=q_{22}$, i.e., $q_{12}=q_{12}^{*}$, the form of the curve χ(K) is independent of the lattice dimension, and the susceptibility takes a finite value at the critical point (Figure 11a,b). With large $\left|q_{12}\right|$, the curve χ(K) does not have a maximum (Figure 11a). It means that the critical exponents $\gamma =\gamma^{\prime }=0$, which agrees with the numbers in Table 2.

If $q_{11}\neq q_{22}$, the susceptibility peak, as expected, is observed at the critical temperature (Figure 11c,d). The height of the peak increases with the lattice dimensions L, which means that the susceptibility diverges at the critical point. This finding agrees with the classical critical exponents $\gamma =\gamma^{\prime }=1$.

3.To obtain the value of the critical exponent *δ*, we need to evaluate the critical temperature $K_{c}$. The temperature dependences of the Binder cumulants [29] for lattices of different dimensions were built for this purpose. The asymptotic value of $K_{c}$ for the case of L→∞ is determined as a point of intersection of these dependences. We define the Binder cumulants for partial and full magnetization in the following way:$$g_{1}=1-\frac{\langle m_{1}^{4}\rangle }{3{\langle m_{1}^{2}\rangle }^{2}},\;g_{2}=1-\frac{\langle m_{2}^{4}\rangle }{3{\langle m_{2}^{2}\rangle }^{2}},\;g=1-\frac{\langle M^{4}\rangle }{3{\langle M^{2}\rangle }^{2}}.$$

When $q_{11}=q_{22}$, in the absence of the external field the full magnetization M is zero. For this reason, we had to use partial magnetization cumulants to determine the critical temperature (Figure 12a). The results obtained with the help of the cumulants $g_{1}$ и $g_{2}$ gave approximately the same value of $K_{c}$. This allowed us to determine the exponent δ=1 to within 10^−3^ (Table 3).

If $q_{11}\neq q_{22}$, the cumulants $g_{1}$ and $g_{2}$ give different estimates of $K_{c}$. At the same time, the cumulants g for the full magnetization do not intersect at one point (Figure 12b). For this reason, we evaluated $K_{c}$ roughly as the middle of the interval where the cumulant g experiences a jump. As can be seen from Table 3, such inaccuracy in the determination of $K_{c}$ results in large deviations of the exponent δ. 

To evaluate the exponent δ, we built relationships between lnM and lnH (Figure 13). 

If $q_{11}=q_{22}$, the relationships between lnM and lnH for lattices of different dimensions L merge into one line already at a sufficiently small H (see Figure 13a). The slope of this relationship was used to evaluate δ. The exponent δ was found to be independent of the ratio $q/\left|q_{12}\right|$, being roughly equal to one (see Table 3). Its deviations from one are about 10^−3^, which is comparable to the error of our evaluation. Thus, the value of the exponent δ in this case fully agrees with its value δ=1 in the mean-field model for $q_{12}=q_{12}^{*}$.

If $q_{11}\neq q_{22}$, the curves lnM as functions of lnH become linear at certain H, whose magnitude decreases with growing L (see Figure 13b). The parameter δ measured by the curve slope differs considerably from the expected value δ=3 in all the investigated configurations (Table 3), and depends on the parameters of the model.
4.Let us consider how the critical exponents change with the increase in the external field in a balanced layered model. We estimated the splitting temperature $K_{s}$ by plotting the temperature dependencies of the following cumulants:$$g_{s}=1-\frac{\langle {\left(m_{1}-m_{2}\right)}^{4}\rangle }{3{\langle {\left(m_{1}-m_{2}\right)}^{2}\rangle }^{2}}.$$The temperature $K_{s}$ was determined as the point of intersection of the cumulants for lattices with various dimensions (Figure 14). We point out that the cumulants $g_{s}$ are usually used for determining the critical temperature of an antiferromagnetic phase transition [30]. The magnetization at the splitting point, $m_{s}$, was calculated as the magnitude of the local maximum of magnetization for the lattice of the maximum size (L=32).

To determine the critical exponents $\beta_{s}$ and $\beta_{s}^{\prime }$, we plotted the dependencies of $\ln(m_{s}-M)$ on $\ln\left|t_{s}\right|$ for the lattices with various dimensions (Figure 15). These curves become linear when $t_{s}<0$, and their slope allows us to estimate the exponent $\beta_{s}$ (see Table 4). For $t_{s}>0$, the dependencies of $\ln(m_{s}-M)$ on $\ln t_{s}$ are nonlinear for the whole range of $t_{s}$, making it impossible to estimate the value of the exponent $\beta_{s}^{\prime }$.

The temperature dependencies of the susceptibility χ near the splitting point are shown in Figure 16. As one can see, when the external field is present, the susceptibility curves depend on the lattice size. The accuracy of our simulations is insufficient to unambiguously attribute this observation to either a divergence of the susceptibility at the temperature $K_{s}$, or to its finite jump.

In order to determine the critical exponents $\delta_{s}$ and $\delta_{s}^{\prime }$, we plotted the dependencies of ln|ΔM| on ln|ΔH| for the lattices with the dimensions ranging from L=12 to L=32 (Figure 17). Independently of the sign of ΔH, these curves have a linear asymptotic, the slope of which allowed us to extract the critical exponents (Table 4).

The results of these simulations are shown in Table 4. The splitting temperature $K_{s}$ differs substantially from the one obtained in the framework of the mean-field model (MFM). The corresponding numbers are presented in column 2 of Table 4. However, when in Equation (33) we use the values of $K_{c}$ and $m_{s}$ obtained as a result of this modelling, we obtain $K_{s}$ rather accurately (compare three numbers in the column 5 of Table 4). The obtained critical exponent of the layered model $\beta_{s}$ takes values in a range of 0.5–0.6, which differs significantly from its value from the mean-field model ($\beta_{s}=1$). The value of the exponent $\beta_{s}$ decreases with the increase in the external field. However, similarly to the mean-field model, the value of $\beta_{s}$ exceeds considerably the values of the exponents of partial magnetization, $\beta_{m_{1,2}}\approx 0.3$, measured in the absence of the external field (in the mean-field model, $\beta_{m_{1,2}}=1/2$). For small values of the external field, the values of the exponents $\delta_{s}$ and $\delta_{s}^{\prime }$ are approximately equal to 1. With the increase in the external field, the exponent $\delta_{s}$ increases, while $\delta_{s}^{\prime }$ decreases.

## 6. Results and Discussions

We have studied the spin system consisting of two antiferromagnetically interacting sub-lattices. We obtained our results using the mean-field model, but we are not able to assert that they hold for the general case. However, the Monte Carlo simulation for the 3D layered model with a finite interaction radius confirms our conclusions. Our results are as follows:In the absence of an external field, the system has a second-order phase transition. If the relationship between the system’s parameters is such that the effective numbers of neighbors in the sub-lattices are equal (12), then the critical exponents of the transition differ significantly from the usual values of the mean-field model (see Table 2), and the susceptibility has a finite value at the critical point. It should be mentioned that the concept of the universality class is still valid because the critical exponents for each sub-lattice remain the same, and the unusual critical behavior takes place only for total values of magnetization and susceptibility.These conclusions are validated by computer simulations of a three-dimensional layered lattice. In particular, the balanced system ($q_{12}=q_{12}^{*}$) is shown to have the following critical exponents: $\gamma =\gamma^{\prime }=0$ and δ=1. The dependences *m*_1_ = *m*_1_(*K*), *m*_2_ = *m*_2_(K) and M=M(K) (Figure 10) generated in the computer simulation reproduce the theoretical curves (Figure 6 and Figure 7).If the condition of equality of the effective number of neighbors (12) is met, the external field does not destroy the phase transition, but only shifts the transition temperature. If the external field is weak enough, $H\leq H_{\max}$, where $H_{\max}$ is defined by (38), the shift of the critical point does not change the qualitative characteristics of the system, and the transition remains a second-order phase transition. In Section 4.2, we show that it is possible to vary the critical temperature over a sufficiently wide range (from $K_{c}$ to $3.3K_{c}$) by varying H within the interval $0\leq H\leq H_{\max}$. Under these conditions, the values of the critical exponents α, γ and δ do not change in the mean-field model, and the exponent β becomes twice as large as its value for the partial magnetizations. In addition, for a finite external field, the system exhibits a jump in the susceptibility at the phase transition point. 

The results of the numerical modeling on a three-dimensional layered lattice demonstrate that the critical exponents β and δ do change when the external field increases (see Table 4). Nevertheless, the results of the numerical modeling coincide with the predictions of the mean-field model in two aspects: (1) the exponent β for the full magnetization considerably exceeds its magnitude for the partial magnetization, and (2) the presence of an external field results in a jump of the susceptibility. 

If $H>H_{\max}$, the shift of the critical point is accompanied by qualitative changes: at the critical point, a first-order phase transition replaces the second-order phase transition, which is accompanied by jumps in the magnetization and the internal energy (Figure 7d).

Since this system has a lot of free parameters, the overall picture of possible regimes is very diverse. Given a certain relation between the parameters of the model and the external field, there may be two first-order phase transitions and two second-order phase transitions (Figure 7f). The general behavior of the temperature dependences of the involved physical quantities, as well as determining the conditions under which various regimes occur, is a complicated problem that deserves to be considered in a separate paper.

2.The behavior of the system depends on the relation between the magnitude of the magnetic field and the critical value $H_{c}$ (11). When $H<H_{c}$, the system’s ground state is an antiferromagnetic state with $m_{1}=-m_{2}$, $\left|m_{1}\right|=\left|m_{2}\right|=1$ and the magnetization $M=\left|p_{1}-p_{2}\right|$. When $H>H_{c}$, the ground state is a ferromagnetic state with $M=m_{1}=m_{2}=1$ (Table 1). Correspondingly, adiabatic cooling of the system (K→∞) transfers it into the antiferromagnetic state at $H<H_{c}$ and into the ferromagnetic state at $H>H_{c}$. The analysis given in Section 4 shows that (except for some special cases) the field dependence of the magnetization M=M(H) at a fixed temperature follows the following pattern:If K is smaller than or of the order of $K_{c}$, then the magnetization M grows monotonously with H, and becomes equal to one when the system reaches the ferromagnetic state at $H\gg H_{c}$; If K is several times greater than $K_{c}$, the picture is different: for $0<H<H_{c}$ there is a plateau of a height $M=\left|p_{1}-p_{2}\right|$, which obtains transferred to the ferromagnetic limit M=1 at $H=H_{c}$ by means of a jump.

The plateau on the curve M=M(H) is shown in Figure 18, where we vary the quantity $\left|p_{1}-p_{2}\right|$ while keeping the other parameters constant. It is seen that the increase in $\left|p_{1}-p_{2}\right|$ results in the growth of the height of the plateau, as well as in the increase in the field magnitude at which the jump to the state M=1 takes place. Note that the curves in Figure 18 resemble the well-known dependence M=M(H) demonstrated by many materials with giant magnetoresistance [31,32,33].

## Figures and Tables

**Figure 1 entropy-25-01428-f001:**
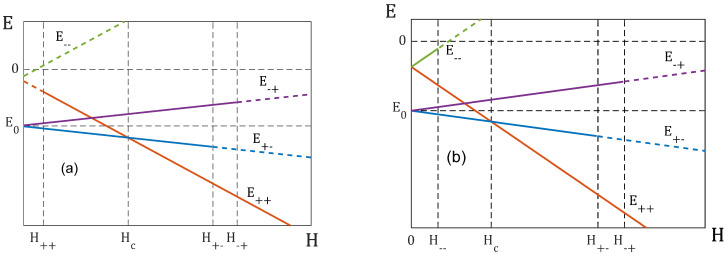
Energies of configurations $S_{++}$, $S_{+-}$, $S_{-+}$ and $S_{--}$ as a function of the strength of the field, H: (**a**) the system does not meet conditions (10); (**b**) the system meets conditions (10). The solid lines mark the range of H, where a configuration is a minimum, the broken lines mark the range where a configuration is not a minimum. Here, we use the following notations: $H_{++}=\max(\left|q_{12}\right|p_{1}-q_{22}p_{2},\left|q_{12}\right|p_{2}-q_{11}p_{1})$, $H_{+-}=q_{22}p_{2}+\left|q_{12}\right|p_{1}$, $H_{--}=\min(q_{22}p_{2}-\left|q_{12}\right|p_{1},q_{11}p_{1}-\left|q_{12}\right|p_{2})$ and $H_{-+}=q_{11}p_{1}+\left|q_{12}\right|p_{2}$.

**Figure 2 entropy-25-01428-f002:**
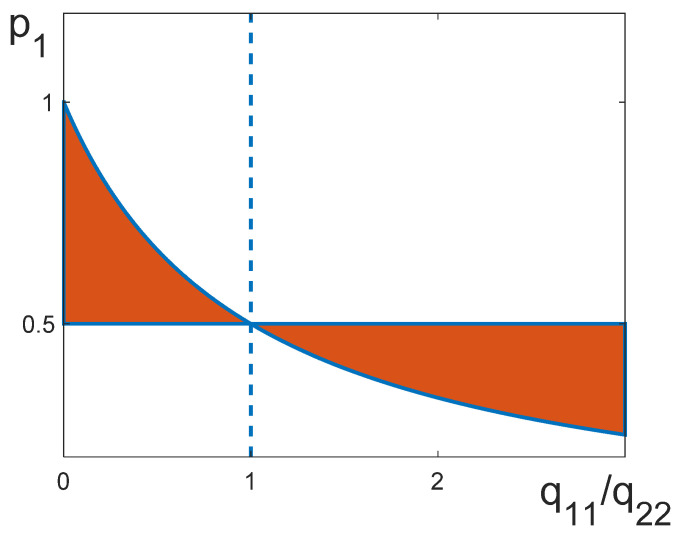
Condition $q_{12}^{*}<0$ is met for parameters lying in the painted area, i.e., between the curve $p_{1}=1/2$ and the curve $p_{1}=q_{22}/(q_{11}+q_{22})$.

**Figure 3 entropy-25-01428-f003:**
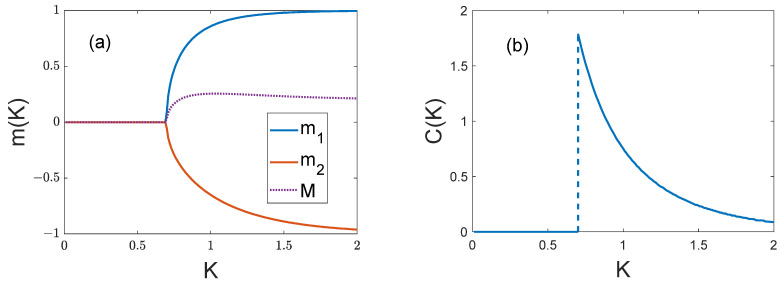
Typical curves for H=0: (**a**) partial magnetizations $m_{1,2}=m_{1,2}(K)$ and full magnetization M=M(K); (**b**) heat capacity C=C(K). The curves are drawn for $q_{11}=2,$ $q_{22}=1,$ $q_{12}=-1,$ $p_{1}=0.6,$ $p_{2}=0.4$.

**Figure 4 entropy-25-01428-f004:**
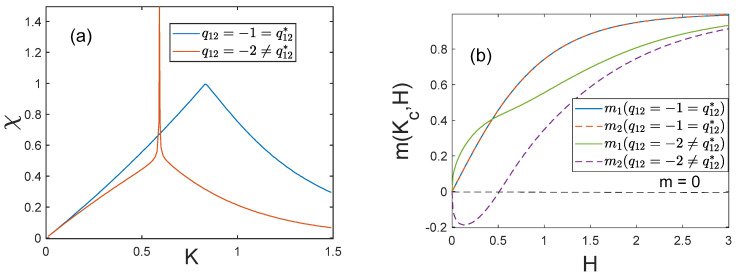
Changes in functions of physical quantities at the “critical” antiferromagnetic interaction: (**a**) susceptibility χ=χ(K,0) at H=0; (**b**) H-dependences of partial magnetizations $m_{1}=m_{1}(K_{c},H)$ and $m_{2}=m_{2}(K_{c},H)$ at the critical temperature. The curves are drawn at $q_{11}=1,$ $q_{22}=2,$ $p_{1}=0.6,$ $p_{2}=0.4$. In panel (**b**), the curves $m_{1,2}=m_{1,2}(K_{c},H)$ merge when $q_{12}=q_{12}^{*}$.

**Figure 5 entropy-25-01428-f005:**
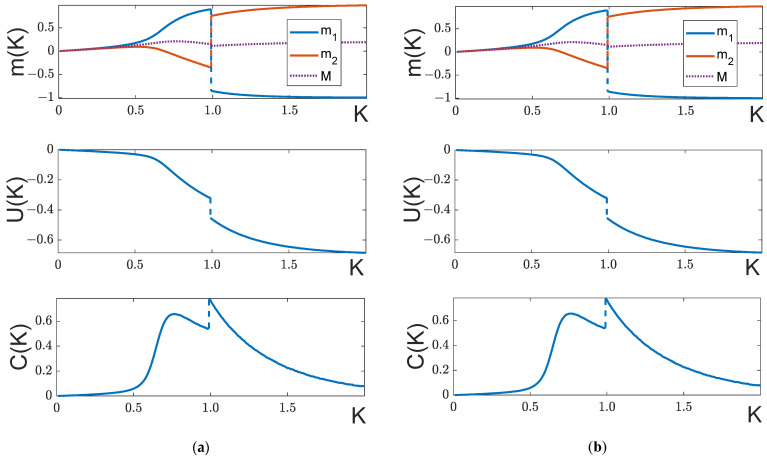
Temperature dependences at $H<H_{c}$. From top to bottom: curves $m_{1,2}(K)$, M(K), U(K) and C(K). (**a**) $q_{11}=2,$ $q_{22}=1,$ $q_{12}=-1,$ $p_{1}=0.6,$ $p_{2}=0.4$, H=0.2 (**b**) $q_{11}=3,$ $q_{22}=1,$ $q_{12}=-1,$ $p_{1}=0.4,$ $p_{2}=0.6$, H=0.2. As we see, there is no phase transition when $p_{1}>p_{2}$, and there is a first-order phase transition when $p_{1}<p_{2}$ (panel (**b**)).

**Figure 6 entropy-25-01428-f006:**
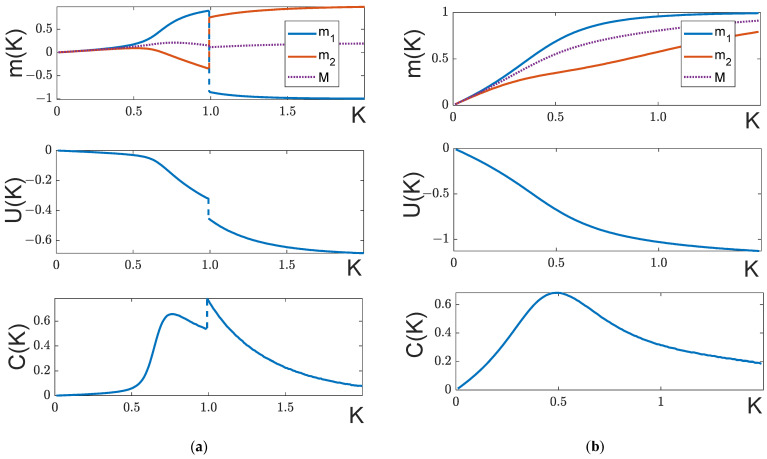
Temperature dependences with $H>H_{c}$. From top to bottom: curves $m_{1,2}(K)$, M(K), и C(K). (**a**) $q_{11}=2,$ $q_{22}=1,$ $q_{12}=-1,$ $p_{1}=0.6,$ $p_{2}=0.4$, H=0.61 (**b**) $q_{11}=1,$ $q_{22}=2,$ $q_{12}=q_{12}^{*}=-1,$ $p_{1}=0.4,$ $p_{2}=0.6$, H=0.5.

**Figure 7 entropy-25-01428-f007:**
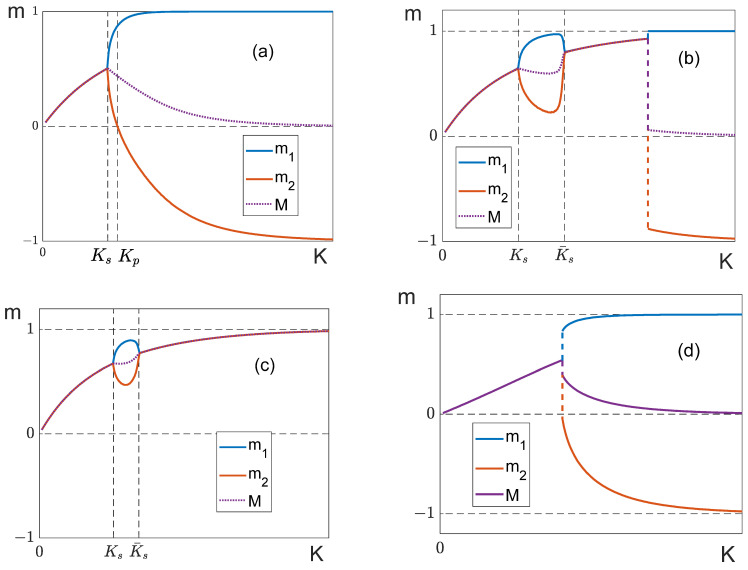
Temperature behavior of magnetizations in the symmetric model. (**a**) $H=0.75H_{c}$; (**b**) $H=0.995H_{c}$; (**c**) $H=1.0075\;H_{c}$; (**d**) $H=0.82H_{c}$; (**e**) $H=1.025H_{c}$; (**f**) $H=\;0.9998H_{c}$. In all cases $\left|q_{12}\right|=2q$, except plots (**d**) $\left|q_{12}\right|=0.75q$ and (**f**) $\left|q_{12}\right|=5q$. q in all plots is equal to 4.

**Figure 8 entropy-25-01428-f008:**
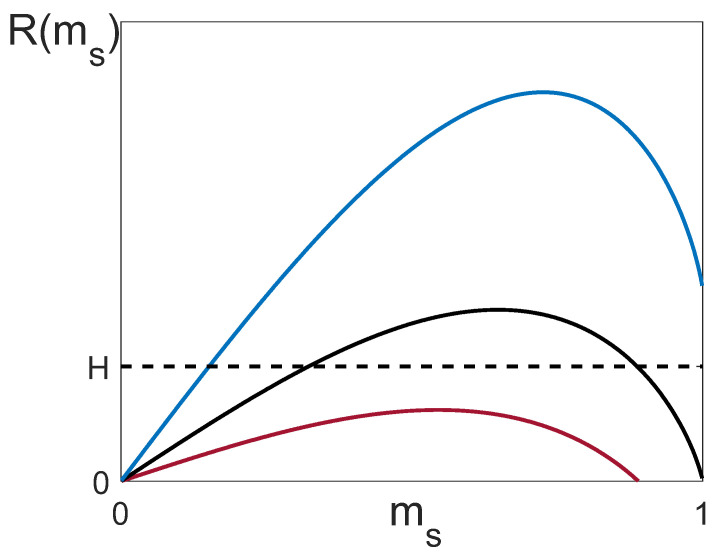
Graphical solution of the equation $R(m^{s})=H$. Depicted is an example when at a given H, Equation (36) has one solution at $\left|q_{12}\right|>q$ (the upper curve), two solutions at $\left|q_{12}\right|=q$ (the middle curve), and no solutions at $\left|q_{12}\right|<q$ (the lower curve).

**Figure 9 entropy-25-01428-f009:**
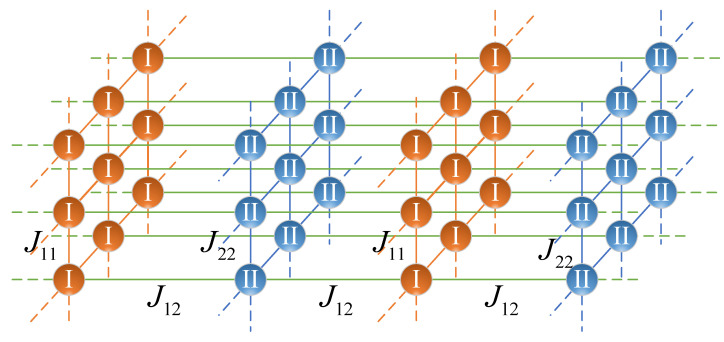
The layered model.

**Figure 10 entropy-25-01428-f010:**
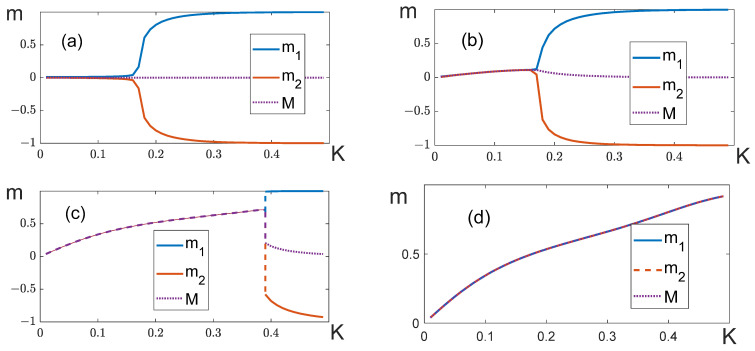
Curves $m_{1}=m_{1}(K)$, $m_{2}=m_{2}(K)$ and M=M(K) at different magnitudes of the external field. (**a**) H=0, (**b**) $H=0.25H_{c},$ (**c**) $H_{c}=0.975H,$ (**d**) $H=0.9975H_{c}$. Everywhere, L=24 and $q_{11}=q_{22}=8,$ $q_{12}=-8$.

**Figure 11 entropy-25-01428-f011:**
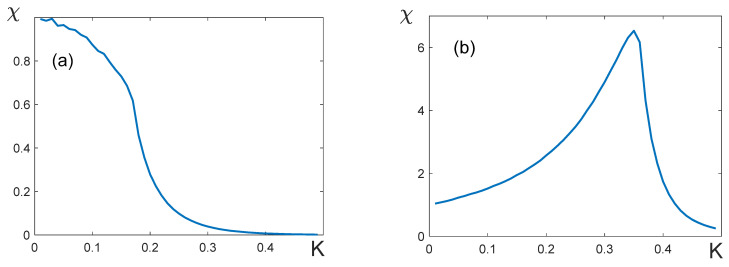
Temperature dependences of susceptibility χ. (**a**) $q_{11}=q_{22}=8,$ $q_{12}=-8$; (**b**) $q_{11}=q_{22}=8,$ $q_{12}=-0.4$; (**c**) $q_{11}=9.6,$ $q_{22}=8,$ $q_{12}=-8$; (**d**) $q_{11}=9.6,$ $q_{22}=8,$ $q_{12}=-0.4$. The form of the curves χ(K) is independent of the lattice dimensions L for (**a**,**b**).

**Figure 12 entropy-25-01428-f012:**
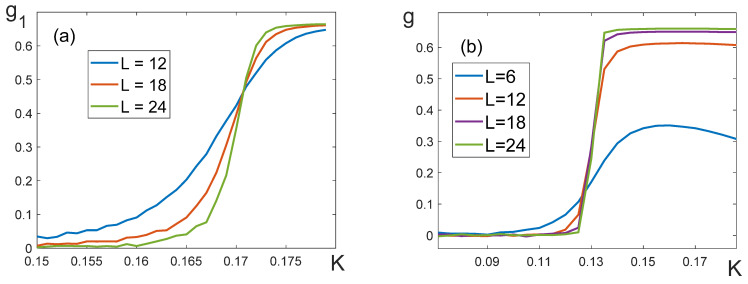
Binder cumulants (**a**) $q_{11}=q_{22}=8,$ $q_{12}=-8$, (**b**) $q_{11}=16,$ $q_{22}=8,$ $q_{12}=-8$.

**Figure 13 entropy-25-01428-f013:**
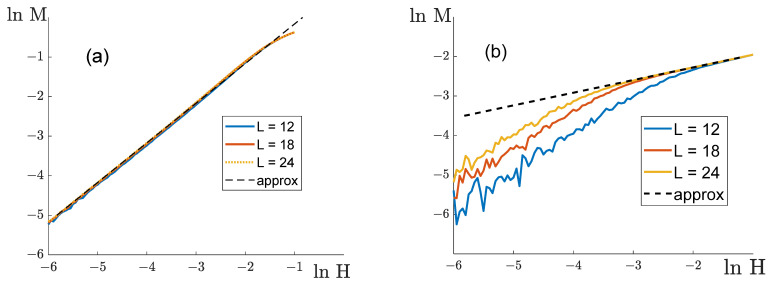
lnM as the function of lnH. The black broken line is a linear approximation used in the calculations of δ. (**a**) $q_{11}=q_{22}=8,$ $q_{12}=-0.4$, which corresponds to $q_{12}=q_{12}^{*}$; (**b**) $q_{11}=24,$ $q_{22}=8,$ $q_{12}=-8$, which corresponds to $q_{12}\neq q_{12}^{*}$.

**Figure 14 entropy-25-01428-f014:**
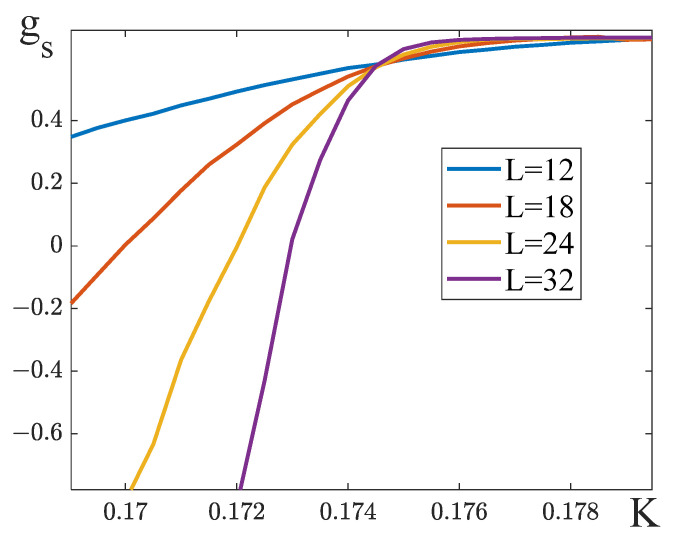
The dependence of the cumulants on $g_{s}$ the reciprocal temperature K. The parameters of the model are: $q_{11}=q_{22}=8,$ $q_{12}=-8$, H=1.

**Figure 15 entropy-25-01428-f015:**
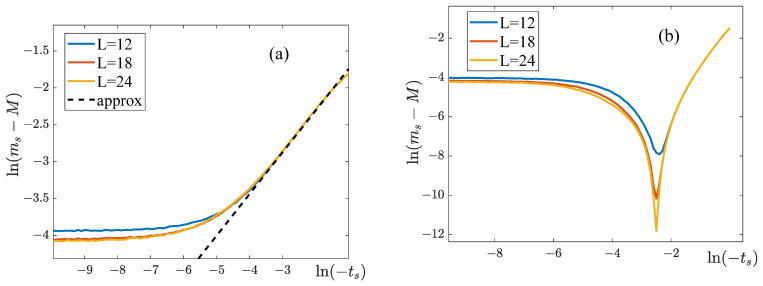
Determining the critical exponent $\beta_{s}$. The parameters of the model are: $q_{11}=q_{22}=8,$ $q_{12}=-8$, H=2. (**a**) Dependencies of $\ln(m_{s}-M)$ on $\ln(-t_{s})$ when $t_{s}<0$. The broken line is our linear approximation, from which we calculated $\beta_{s}$. (**b**) Dependencies of $\ln(m_{s}-M)$ on $\ln t_{s}$ when $t_{s}>0$.

**Figure 16 entropy-25-01428-f016:**
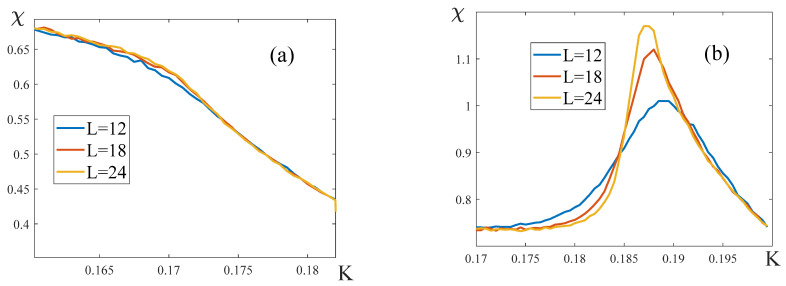
Temperature dependencies of the susceptibility near the critical point. The parameters of the model are: $q_{11}=q_{22}=8,$ $q_{12}=-8$. The magnitude of the external field is (**a**) H=0.1, (**b**) H=2.

**Figure 17 entropy-25-01428-f017:**
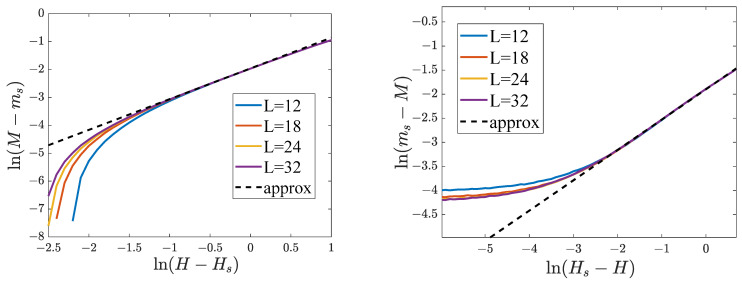
Determining the critical exponents $\delta_{s}$ and $\delta_{s}^{\prime }$. The parameters of the model are: $q_{11}=q_{22}=8,$ $q_{12}=-8$, $H_{s}=2$.

**Figure 18 entropy-25-01428-f018:**
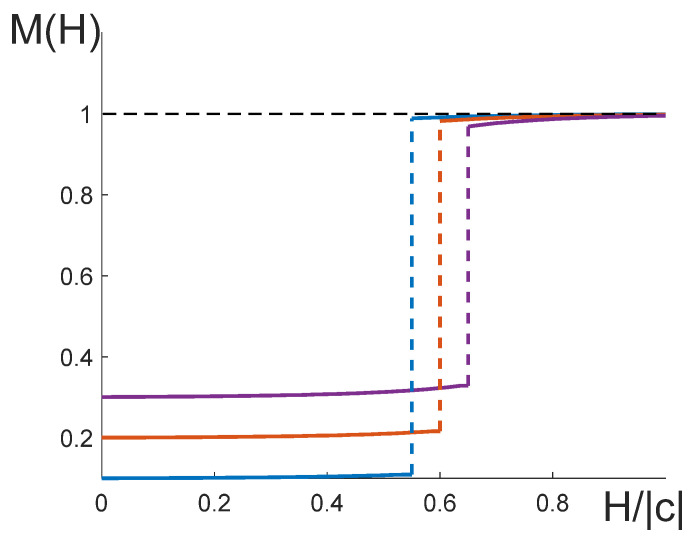
Magnetization versus the field H: from bottom to top, $p_{1}-p_{2}=0.1,\;0.2,\;0.3$. The curves are plotted for the case of $K=4K_{c}$ with $q_{11}=4.4$, $q_{22}=4$, $q_{12}=-2$.

**Table 1 entropy-25-01428-t001:** Ground and metastable states in the zero-temperature limit (K→∞).

Condition	Ground State (Global Minimum)	Metastable State (Local Minimum)
$H<H_{c},\;p_{1}>p_{2}$	$m_{1}=1,\;\;\;m_{2}=-1,\;M=p_{1}-p_{2}$	$m_{1}=1,\;\;\;m_{2}=1,\;\;M=1$
$H<H_{c},\;p_{1}<p_{2}$	$m_{1}=-1,\;m_{2}=1,\;\;M=p_{2}-p_{1}$	$m_{1}=1,\;\;\;m_{2}=1,\;\;M=1$
$H>H_{c},\;p_{1}>p_{2}$	$m_{1}=1,\;\;\;m_{2}=1,\;\;M=1$	$m_{1}=1,\;\;\;m_{2}=-1,\;M=p_{1}-p_{2}$
$H>H_{c},\;p_{1}<p_{2}$	$m_{1}=1,\;\;\;m_{2}=1,\;\;M=1$	$m_{1}=-1,\;m_{2}=1,\;\;M=p_{2}-p_{1}$

**Table 2 entropy-25-01428-t002:** Critical exponents.

Critical Exponent	$q_{11}p_{1}+q_{12}p_{2}\neq q_{22}p_{2}+q_{12}p_{1}$ $(q_{12}\neq q_{12}^{*}$)	$q_{11}p_{1}+q_{12}p_{2}=q_{22}p_{2}+q_{12}p_{1}$ $(q_{12}=q_{12}^{*}$)
α	0	0
β	1/2	3/2
$\gamma =\gamma^{\prime }$	1	0
δ	3	1
Scaling hypothesis	confirmed	?

**Table 3 entropy-25-01428-t003:** Evaluation of the exponent δ for the layered model.

Model Parameters	$K_{c}$	δ
Balanced system ($q_{12}=q_{12}^{*}$)		
$q_{11}=q_{22}=8,\;q_{12}=-0.4$	0.3580	0.9982
$q_{11}=q_{22}=8,$ $q_{12}=-2$	0.2710	0.9921
$q_{11}=q_{22}=8,$ $q_{12}=-4$	0.2217	0.9974
$q_{11}=q_{22}=8,$ $q_{12}=-8$	0.1707	1.0085
$q_{11}=q_{22}=8,$ $q_{12}=-12$	0.1430	1.0012
Unbalanced system ($q_{12}\neq q_{12}^{*}$)		
$q_{11}=4,$ $q_{22}=8,$ $q_{12}=-8$	0.1985	2.2929
$q_{11}=16,$ $q_{22}=8,$ $q_{12}=-4$	0.1615	4.4033
$q_{11}=16,$ $q_{22}=8,$ $q_{12}=-8$	0.1310	2.5688
$q_{11}=16,$ $q_{22}=8,$ $q_{12}=-12$	0.1125	3.1631
$q_{11}=24,$ $q_{22}=8,$ $q_{12}=-8$	0.1010	3.1175

**Table 4 entropy-25-01428-t004:** Critical exponents for the layered model in the splitting point ($q_{11}=q_{22}=8$).

$q_{12}$	$K_{c}$/MFM	$\beta_{m_{1,2}}$	$H_{s}$	$K_{s}$/MFM/Equation (33)	$m_{s}$/MFM	$\beta_{s}$	$\delta_{s}$ $/\delta_{s}^{\prime }$
−4	0.2217/0.1667	0.2826	0.5	0.2257/0.1694/0.2246	0.1145/0.1271	0.6106	1.0688/0.8623
			1	0.2387/0.1798/0.2350	0.2381/0.2700	0.5814	1.0474/0.6352
−8	0.1707/0.1250	0.3128	0.1	0.1707/0.1250/0.1707	0.0110/0.0125	0.6077	1.0425/1.0502
			0.5	0.1717/0.1255/0.1712	0.0550/0.0627	0.5922	1.0703/1.0083
			1	0.1744/0.1270/0.1728	0.1105/0.1263	0.5814	1.1014/0.7923
			2	0.1866/0.1342/0.1801	0.2287/0.2622	0.5650	1.1015/0.6304
−16	0.1240/0.0833	0.3032	2	0.1271/0.0847/0.1254	0.1051/0.1260	0.5049	1.1609/0.7473
			4	0.1383/0.0893/0.1301	0.2168/0.2588	0.4906	1.1924/0.5753

## Data Availability

The data presented in this study are openly available in FigShare at https://doi.org/10.6084/m9.figshare.24218805.v1.

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
