# Peer review of "Characteristics of an Ising-like Model with Ferromagnetic and Antiferromagnetic Interactions"

_entropy, 2023, doi:10.3390/e25101428_

Round 1
Reviewer 1 Report
The paper is containing an impressive work of analysis on the condensed matter system of spins such as e.g. a laired structure of ferromagnetic spins. To calculate the system is used a mean-field approximation and then using numerical simulation of the system as is done but require complicated boundary conditions as the authors also hint. I recommend the paper for publication in Entropy.
Author Response
Dear Reviewer 1,
we are glad that you estimate our paper positively.
Thank you for your attention to the paper.
Leonid Litinskii,
correspondence author of paper by B. Kryzhanovsky, Vl. Egorov and L. Litinskii Characteristics of an Ising-like Model with Ferromagnetic and Antiferromagnetic Interactions.
Reviewer 2 Report
This is a nice paper dealing with the investigation of the critical properties of a spin system, by means of the mean-field approximation, consisting of two interacting sub-ensembles having an antiferromagnetic inter-ensemble interaction. In each sub-ensemble the intra-ensemble interaction is ferromagnetic. It is shown that, if the two sub-systems have the same effective number of nearest neighbors, the set of critical exponents is not in the classical form of the Ising model treated in the framework of mean-field theory but in the non-classical form with different values for the exponents related to the magnetization vs. temperature and to the external magnetic field as a function of the magnetization on the critical isotherm. It is also demonstrated that the system undergoes two second-order phase transitions and two first-order phase transitions. The peculiarity of the system investigated is to have not only the global minimum of energy related to the antiferromagnetic order but also the local minimum of energy marking the ferromagnetic order in the presence of an antiferromagnetic interaction. This is a really interesting aspect affecting the underlying physics of these two antiferromagnetically coupled subsystems and is certainly worth of investigation.
For all the above, in my opinion, the choices of the journal and of the special issue are appropriate. Moreover, the planning of the paper is very good and the main arguments are clearly outlined. For these reasons, certainly this article could merit publication in Entropy.
However, prior to the publication, the authors should address the following points and comments:
1. I understand that an exact calculation of the critical exponents by means of renormalization group method or transfer matrix method would be a complicated task for the system under study and the mean-field approximation is a good choice to simplify the calculations. It is known that the exact critical exponents are well reproduced by the mean-field approximation at least in 3D systems but even better in 4D systems. The authors confirm the mean-field results by means of Monte Carlo simulations but I expect that also in the simulations some approximations have been done. On this basis, do the authors think that the analysis done is realistic for the complicated system under study? What could be the possible limits of this treatment? A brief discussion of the advantages and disadvantages in the introductory part would be helpful for a reader.
2. Is the concept of universality class still valid if it is enough to change the number of effective nearest neighbors to cause a change of the numerical values of two typical mean-field critical exponents belonging to the Ising model universality class? I understand that the general inequalities among critical exponents still hold and this is proved here.
3. In relation to point 2. What do the authors mean for non-classical form of the critical exponents? Isn’t true that the critical exponents belonging to the same class of universality of the Ising model would be the same when calculated in the framework of the mean field approximation? Why two of them change?
4. Figure 1 is a key figure showing the energies of the different configuration vs. the amplitude of the magnetic field but, in my opinion, it is not of immediate comprehension due to the high number of lines. I appreciate the authors’ efforts in explaining part of it in the caption but I would suggest to give more details also in the main text.
5. I understand that the discussion is related to thermodynamics and thermodynamic quantities such as the calculation of the specific heat. However, I didn’t find any discussion or part related to the entropy calculation. It would be enough that the authors include a brief discussion on the qualitative behavior of the entropy of the spin system under study.
6. It is stated that the external magnetic field does not destroy the phase transitions but only shifts the critical points. This result is obtained because the analysis is done in the framework of the mean-field theory. In an exact treatment, the presence of an external field suppresses critical points and the occurrence of a second-order phase transition. Is this the case? If yes, what is the reasonable physics which can be extracted from this analysis?
7. Even though for the authors could be obvious, some details of the steps leading to the mean-field free energy for spin of Eq. (6) would be useful. What are explicitly the terms with magnetization close to equilibrium in the sum of Eq.(4) ?
8. I think that a more detailed discussion of the results summarized in the key Table 1 summarizing ground and metastable states in the zero-temperature limit would strengthen the message conveyed by this study.
9. The authors should not ignore some papers dealing with magnetic systems described also by the Ising-1 model in some way similar to the system investigated here also in relation to the types of phase transitions occurring and certainly belonging to the same universality class. See, e.g., 1) Zhang Q, Wei G-Z, and Liang Y -Q. Phase diagrams and tricritical behavior in spin-1 Ising model with biaxial crystal-field on honeycomb lattice. J. Magn. Magn. Mat. 2002 253, 45–50; 2) Ez-Zahraouy H and Kassou-Ou-Ali A. Phase diagrams of the spin-1 Blume-Capel film with an alternating crystal field. Phys. Rev. B 69 2004, 064415; 3) Zivieri R. Critical behavior of the classical spin-1 Ising model for magnetic systems. AIP Advances 2022 12, 035326; 4) Kaneyoshi T. Contribution to the theory of spin-1 Ising models. J. Phys. Soc. Japan 56, 1987 933–941; 5) Kaneyoshi T. The phase transition of the spin-one Ising model with a random crystal field. J. Phys. C: Solid State Phys. 1988 21, L679–L682.
These papers could give more generality to this study and could be briefly discussed in the Introduction.
Minor points:
Abstract, line 9: Please, rephrase the sentence, it is not clear
Line 57: “gigantic magnetoresistance” should be replaced by “giant magnetoresistance”
c Line 72: N the total number of spins should be defined.
Eq. (2) is the general Hamiltonian and not really the mean-field approximation for the Hamiltonian as instead is stated. The mean field expression of the Hamiltonian is Eq.(3).
e Caption of Fig.1: it would be better to replace “line” with “lines”
In my opinion, English Language is good. The authors should sometime rephrase some sentences starting from the Abstract.
Author Response

(The authors gave the same response as above.)

Round 2
Reviewer 2 Report
The authors have addressed all my previous comments and remarks and have made the amendments in the manuscript accordingly. I understand that the problem of entropy will be addressed in a further study. However, the thermodynamics of the system is studied and this is enough to consider this work suitable for the journal.
I believe that this study has the merit to shed light on the statistical physics of Ising-like models in the presence of both ferromagnetic and antiferromagnetic coupling and strongly advances this topic. For all these reasons it certainly deserves publication in the journal.